# Cardiac myosin binding protein-C phosphorylation as a function of multiple protein kinase and phosphatase activities

Thomas Kampourakis [1], Saraswathi Ponnam[1], Kenneth S. Campbell [2], Austin Wellette-Hunsucker [2] & Daniel Koch [3] ✉

Phosphorylation of cardiac myosin binding protein-C (cMyBP-C) is a determinant of cardiac myofilament function. Although cMyBP-C phosphorylation by various protein kinases has been extensively studied, the influence of protein phosphatases on cMyBP-C's multiple phosphorylation sites has remained largely obscure. Here we provide a detailed biochemical characterization of cMyBP-C dephosphorylation by protein phosphatases 1 and 2 A (PP1 and PP2A), and develop an integrated kinetic model for cMyBP-C phosphorylation using data for both PP1, PP2A and various protein kinases known to phosphorylate cMyBP-C. We find strong site-specificity and a hierarchical mechanism for both phosphatases, proceeding in the opposite direction of sequential phosphorylation by potein kinase A. The model is consistent with published data from human patients and predicts complex non-linear cMyBP-C phosphorylation patterns that are validated experimentally. Our results suggest non-redundant roles for PP1 and PP2A under both physiological and heart failure conditions, and emphasize the importance of phosphatases for cMyBP-C regulation.

The cardiac contractile function is regulated by several coordinated and periodic processes, including electrical excitation and repolarization, calcium cycling, and thick and thin filament activation. Under stress conditions, humoral and neuronal signals can improve cardiac performance by changing key parameters of these processes, thereby increasing heart rate (chronotropy), force generation (inotropy) and relaxation (lusitropy). Pathological conditions can impair these regulatory mechanisms, leading to heart disease and heart failure (HF). Often, however, it is not clear whether associated molecular changes are the cause or effect of the dysfunction at the cellular or organ level (or, in fact, both).

As a component of the sarcomere (Fig. 1a, left), the cardiac isoform of myosin-binding protein-C (cMyBP-C) is involved in regulating heart muscle contractility via binding of its N-terminal domains to the actin-containing thin and myosin-containing thick filaments,

controlling their regulatory states[1–5]. Thin filament binding has been associated with an activating effect on contractility by increasing its calcium sensitivity, whereas thick filament binding stabilizes the myosin heads OFF-state and reduces contractility. Physiologically, the interaction of cMyBP-C with both filament systems is controlled by phosphorylation at several sites in the cardiac-specific m-motif by protein kinases[6,7] (Fig. 1a, right). The relevance of cMyBP-C for normal heart muscle performance is further emphasized by studies showing that ~50% of patients suffering from familial hypertrophic cardiomyopathy (HCM) carry mutations in the gene encoding for cMyBP-C[8]. In addition, cMyBP-C phosphorylation is found to be strongly reduced during HF, potentially being both a cause and effect of cardiac dysfunction[9]. Consistently, ablation of cMyBP-C phosphorylation leads to cardiomyopathy and HF in transgenic animal models[10,11].

[1]Randall Centre for Cell and Molecular Biophysics; and British Heart Foundation Centre of Research Excellence, King's College London, London SE1 1UL, United Kingdom. [2]Division of Cardiovascular Medicine, University of Kentucky, Lexington, KY, USA. [3]Max Planck Institute for Neurobiology of Behavior—caesar, Ludwig-Erhard-Allee 2, 53175 Bonn, Germany. ✉e-mail: dkoch.research@protonmail.com

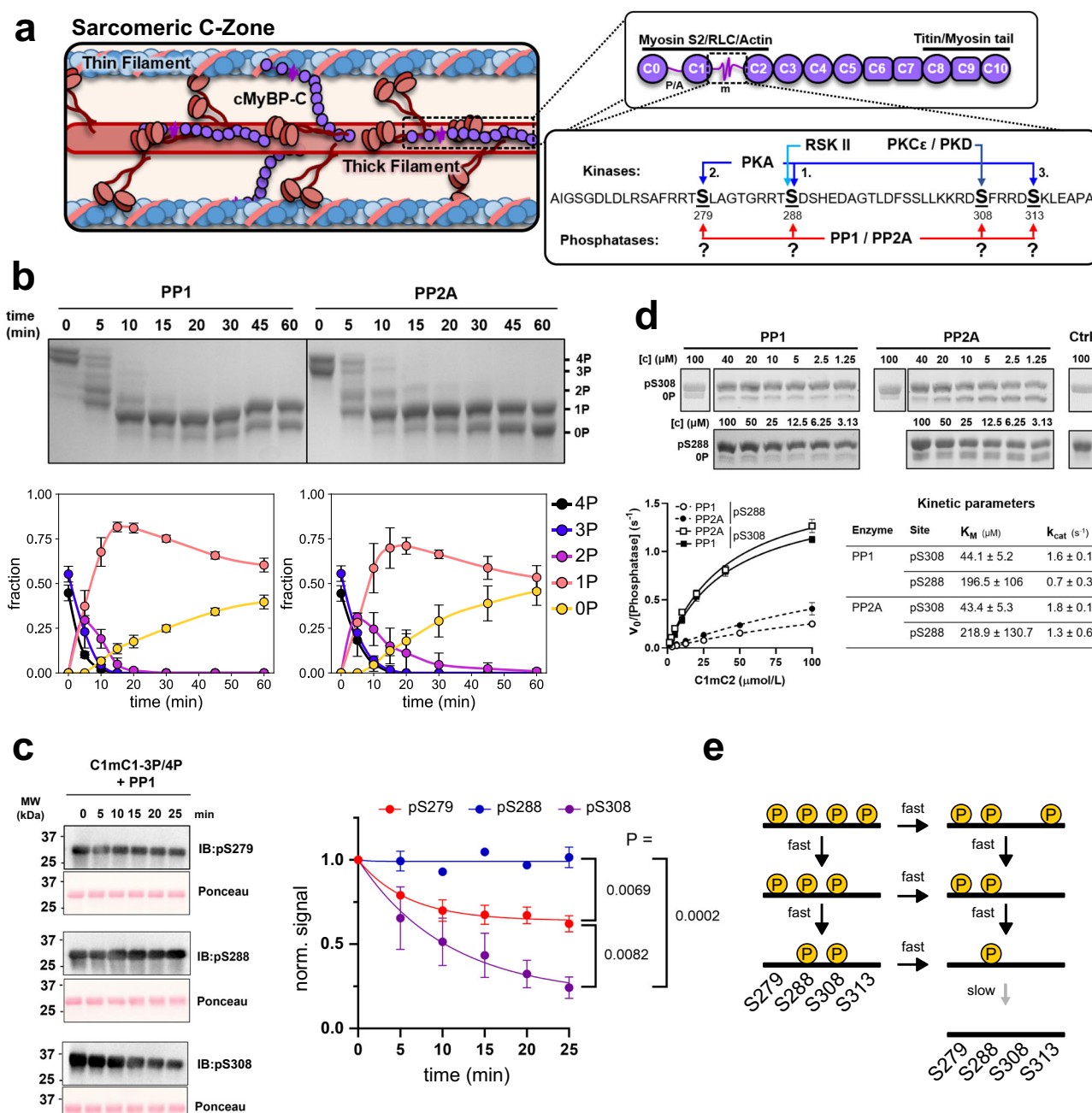

**Fig. 1 | cMyBP-C dephosphorylation by PP1 and PP2A. a** left and top right: cMyBP-C is located in the C-zone of the sarcomere. It is composed of eight Ig-, three FnIII-domains, a P/A linker region between C0 and C1, and the semi-structured m-motif between domains C1 and C2, which features multiple phosphorylation sites. The C-terminus is integrated into the thick filament, whereas the N-terminus can bind both to the thin filament, stabilizing the ON-state, and to the thick filament, stabilizing its OFF-state. The phosphorylation states of the m-motif between cMyBP-C C1 and C2 regulate these interactions in a site-specific manner. Bottom right: cMyBP-C can be phosphorylated by multiple kinases, including PKA, CaMKII (not shown), PKCε, and PKD, in a site-specific manner. While both PP1 and PP2A have been reported to dephosphorylate cMyBP-C, the reaction mechanism, site-specificity, and kinetics of these reactions are unknown. **b** 20 μmol/L 3P/4P-C1mC2 (~50% pS279, pS288, pS308, and ~50% pS279, pS288, pS308, pS313) were dephoshorylated by 100 nmol/L PP1 or PP2A for 60 min. Samples were taken at different time points and analyzed by PhosTag™ -SDS-PAGE. Mean ± SD from n = 3 experiments. **c** left: Immunoblot analysis of 3P/4P-C1mC2 dephosphorylation by PP1 (10 μM 3P/4P-C1mC2, 50 nmol/L PP1) using pS279-, pS288- and pS308-cMyBP-C specific antibodies. Right: quantified mean ± SEM from n = 3 experiments analyzed via 1-way ANOVA followed by Tukey's multiple comparison test. **d** Michaelis–Menten kinetics for dephosphorylation of pS288- and pS308-C1mC2 by PP1 and PP2A. After the addition of phosphatase (100 nmol/L), reactions were quenched after 2 min (pS308) or 10 min (pS288), and the amount of dephosphorylated product quantified via PhosTag™ -SDS-PAGE. Datapoints are mean ± SEM from n = 3 experiments. See Methods for details. **e** Proposed reaction scheme for dephosphorylation of cMyBP-C by PP1 and PP2A. pS279, pS288, and pS313 sites are dephosphorylated sequentially in the reverse order in which they are added by PKA, whereas pS308 can independently be dephosphorylated at any point.

A major obstacle to understanding the role of cMyBP-C is posed by the multitude of phosphorylation sites and the functional discrepancy between cMyBP-C phosphorylation and phosphomimetic models, making interpretation of the phenotype of genetic rodent models with serine-to-aspartate substitutions difficult[12–14]. This is further complicated by the multiplicity of enzymes involved in regulating these sites, including protein kinases A (PKA), protein kinase C (PKC), protein kinase D (PKD), ribosomal S6 kinase (RSK2), Ca²⁺/calmodulin-

dependent protein kinase II (CaMKII) and glycogen-synthetase kinase 3 (GSK3) (reviewed in[7]). Phosphorylation of cMyBP-C by PKA at pS279, pS288, and pS313 (*rattus norvegicus* sequence) leads to both enhanced inotropy and lusitropy, making cMyBP-C an important mediator of β-adrenergic stimulation on the level of the contractile myofilaments[15–17]. We have previously shown that phosphorylation of cMyBP-C by PKA proceeds in a strictly sequential fashion (pS288→pS279→pS313), whereas RSK2 and PKCε/PKD phosphorylate cMyBP-C site-specifically at S288 and S308, respectively[6].

Generally, kinases often receive more attention and are studied in greater detail than phosphatases, despite the latter being interesting therapeutic targets[18–20]. Although the function and regulation of protein phosphatases are not well understood compared to the corresponding kinases, the output of any phosphorylation/dephosphorylation cycle equally depends on the joint activity of kinases and phosphatases, i.e., the phosphorylation state of any protein needs to be considered as a function of both enzyme classes[21]. In the case of multisite and multi-enzyme phosphorylation systems, the relationship between enzyme activity and phosphorylation states is often far from trivial: Non-linearity and reciprocal relationships can give rise to complex behavior such as switch-like transitions, oscillations or multistability[22–25]. Understanding such system behavior and its physiological implications requires quantitative approaches and knowledge of both the kinetics and mechanisms of the involved reaction steps[25].

While significant progress has been made in determining the identity and site-specificity of the kinases regulating cMyBP-C[6,7], little is known about how cMyBP-C is dephosphorylated. Previous studies reported that cMyBP-C can be dephosphorylated by both protein phosphatase 1 (PP1) and protein phosphatase 2 A (PP2A) in vitro[26] and in vivo[27–29], but it is unknown what the site-specificity, mechanism, and kinetics of these dephosphorylation reactions are, prohibiting an integrated quantitative understanding of cMyBP-C phosphorylation state regulation.

Aiming to fill this gap, this study provides a detailed biochemical characterization of cMyBP-C dephosphorylation by both PP1 and PP2A, and introduces a kinetic model for cMyBP-C phosphorylation that integrates the activities of both kinases (PKA, RSK2, PKC) and phosphatases (PP1 and PP2A). Although other kinases (e.g., CamKII, PKD) are relevant for cMyBP-C phosphostate regulation, the five enzymes considered here are involved in the regulation of all phosphorylation sites in the m-motif and thus represent a good starting point. Our model is trained using data from >40 kinetic experiments and validated by independent biochemical experiments. Our findings show that PP1 and PP2A follow a strongly sequential mechanism for cMyBP-C dephosphorylation that likely depends on intrinsic structural transitions in the m-motif itself. Simulations and experiments revealed site-specific and non-linear cMyBP-C phosphorylation response curves with switch-like responses for higher phosphorylated forms. Furthermore, the model reveals pronounced differences between the effects of PP1 and PP2A in the presence of PKC. We show that our model is consistent with data on cMyBP-C phosphorylation from human HF patients but that the observed changes require complex remodeling of the signaling activities in addition to β-adrenergic receptor desensitization and increased total phosphatase activity, e.g. changes in the PP1/PP2A ratio. Our findings suggest non-redundant roles for PP1 and PP2A that will help to better disentangle the physiological effects of cMyBP-C phosphorylation and may have important implications for the modulation of cMyBP-C phosphorylation during HF.

## Results

### PP1 and PP2A dephosphorylate cMyBP-C in reverse order of PKA phosphorylation

To elucidate how and at which sites PP1 and PP2A can dephosphorylate cMyBP-C, we dephosphorylated an almost completely phosphorylated

fragment of cMyBP-C containing the m-motif flanked by domains C1 and C2 (C1mC2) (mixture of ~50% pS279, pS288, pS308 and ~50% pS279, pS288, pS308, pS313) with both protein phosphatases. After an initial phase of rapid dephosphorylation of the tetrakis-, tris- and bis-phosphorylated species by both PP1 and PP2A, dephosphorylation slowed down for the remaining mono-phosphorylated C1mC2 (Fig. 1b). However, the accumulation of completely dephosphorylated C1mC2 indicates that both phosphatases can dephosphorylate C1mC2 at all sites. Judging from the slow decline of 1P cMyBP-C at >20 min, however, at least one site appears to be dephosphorylated notably slower than others.

Next, we set out to explore the mechanism and kinetics for the dephosphorylation of the individual sites. Utilizing site-specific antibodies to analyze the dephosphorylation of C1mC2 by PP1 we found the pS308 signal to decline first, followed by a decrease in pS279, whereas the pS288 signal remained constant, indicating that dephosphorylation of pS288 likely accounts for the slow dephosphorylation phase (Fig. 1c). We used Michaelis-Menten kinetics to characterize dephosphorylation of the mono-phosphorylated substrates pS308- and pS288-C1mC2. We found that only pS288 is a poor substrate for either phosphatases, exhibiting an about four-fold higher $K_m$ and lower $k_{cat}$ values compared to pS308. (Fig. 1d).

This suggests that in the fully phosphorylated C1mC2 fragment, pS308 and pS313 are dephosphorylated first, leaving only pS279 and pS288. Analyzing the identity of the phosphorylation sites in the bis-phosphorylated species of the dephosphorylation timecourses using mass spectrometry confirmed that for both PP1 and PP2A, only pS279 and pS288 are reliably detectable (Supplementary Fig. 1).

Taken together with previously published results localizing the phosphorylation sites in C1mC2[6], we propose the following reaction mechanism for cMyBP-C dephosphorylation by both PP1 and PP2A (Fig. 1e): As pS313 is dephosphorylated before pS279 and pS288, and since pS288 is dephosphorylated very slowly, dephosphorylation of the PKA sites in the absence of pS308 likely occurs predominantly by following the sequence pS313→pS279→pS288, i.e. in reverse sequence of PKA phosphorylation. While it is possible that some pS288 sites may be removed before pS279, the slow kinetics of pS288 dephosphorylation will make this path less dominant. pS308 is dephosphorylated before pS279 and pS288 in the 3P/4P-C1mC2 dephosphorylation experiments but can also be dephosphorylated efficiently in the absence of other sites. We conclude that pS308 can likely be removed by PP1/PP2A at any point, which is biologically plausible for a site that can be independently regulated by multiple kinases of the PKC/PKD family[30,31].

### An integrated quantitative model of cMyBP-C phosphorylation state regulation

Next, we developed an integrated mathematical model of cMyBP-C phosphorylation considering both protein kinase and phosphatase activities. Such an approach is particularly important for multisite phosphorylation systems such as cMyBP-C, as quantitative differences and cross-talk between sites or enzymes often lead to the emergence of non-linear behaviors such as ultrasensitivity or multistability, with important consequences for cellular information processing[25].

For simplicity we introduce a new nomenclature that makes referencing cMyBP-C phosphorylation states more convenient and is independent of amino acid numbering across species (Fig. 2a): we denote rat pS288, pS279 and pS313 by α, β and γ, reflecting the sequential order of their addition by PKA[6], and the PKCε/PKD site pS308 as δ.

Based on our mechanistic insights on cMyBP-C (de-)phosphorylation provided here and in earlier studies[6], it is likely that cMyBP-C can adopt eight distinct phosphorylation states as a result of different kinase and phosphatase activities (Fig. 2b). To keep the model simple and limit the number of parameters, reactions were described by

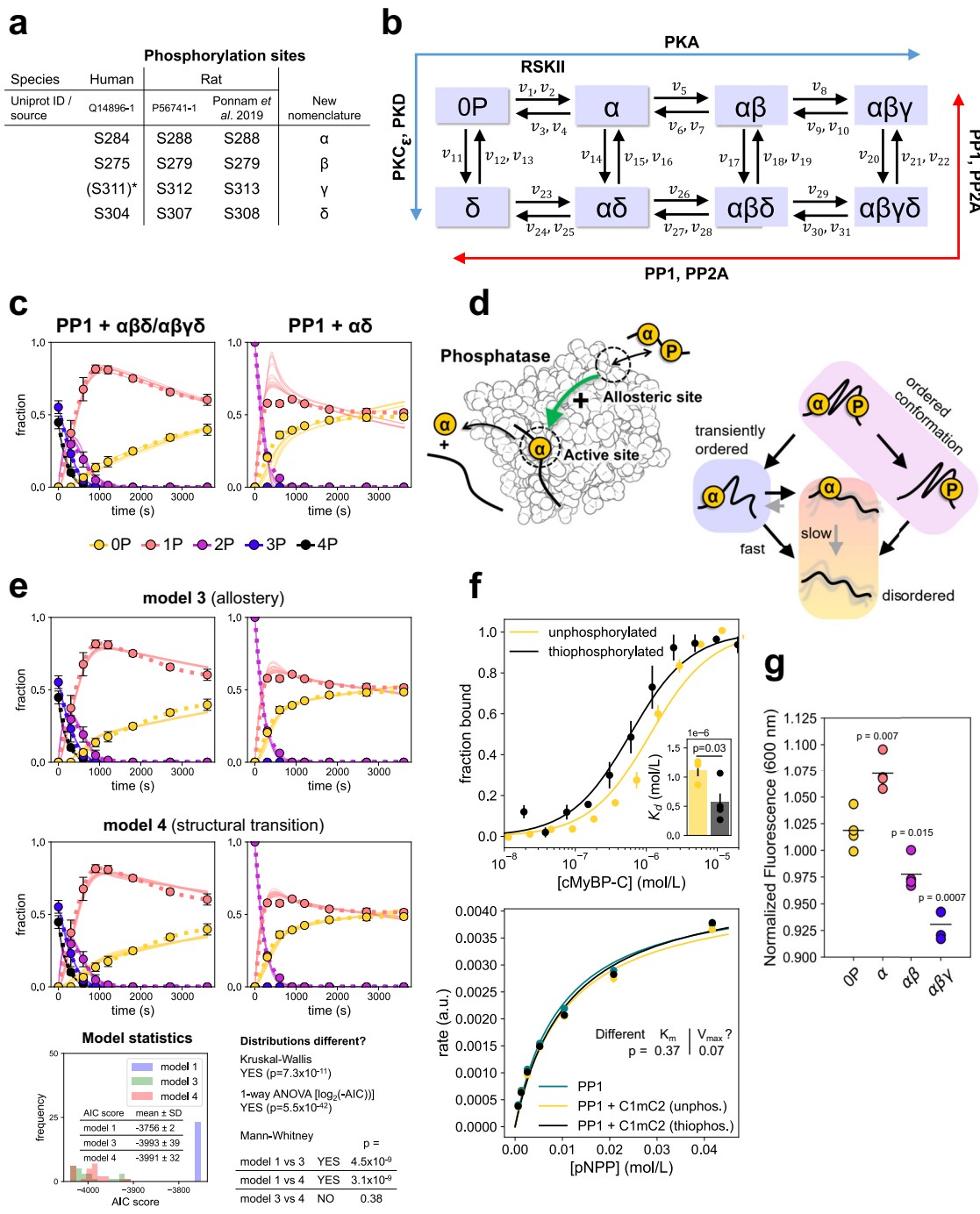

**Fig. 2 | A quantitative model for cMyBP-C phosphostate regulation. a** New nomenclature for cMyBP-C-phosphorylation sites. *Phosphorylation in humans not confirmed. **b** Reaction scheme of proposed cMyBP-C-phosphorylation state model integrating the activities of multiple kinases and phosphatases. **c** Experimental dephosphorylation time course data and model fits showing good agreement for 3P/4P-cMyBP-C dephosphorylation and systematic discrepancies for 2P-cMyBP-C dephosphorylation (here: αδ). Thin lines are model fits ($n = 14$ parameter sets), dots are the experimental data (mean ± SD from $n = 3$ (left)/$n = 2$ (right)). **d** Hypothetical mechanisms based on allosteric activation of phosphatases (model 3) or structural transitions (model 4) in cMyBP-C to explain the abrupt decline in 1P dephosphorylation rate after 2P-cMyBP-C depletion. **e** Top four panels: improved model fits the same experimental data as in c following implementation of the allostery ($n = 24$ parameter sets) or structural transition ($n = 25$ parameter sets) hypotheses

into the model. Bottom panel: model statistics of model 1 (Michaelis-Menten–kinetics only), 3, and 4 for PP1 data. Model performance according to the Akaike information criterion (Krude-Wallis, 1-way ANOVA and Mann-Whitney $U$ tests corrected for multiple comparisons with the Benjamini-Hochberg procedure with a false discovery rate of 0.05). **f** Top: binding of unphosphorylated ($n = 3$) and thiophosphorylated ($n = 4$) C1mC2 to PP1 as measured by Microscale Thermophoresis (mean ± SEM). Inset: $K_d$ values compared via Welch's t-test. Bottom: pNPP dephosphorylation kinetics of PP1 in the absence or presence of unphosphorylated or thiophosphorylated C1mC2 ($n = 3$, mean ± SEM). **g** Peak SYPRO-signal at 600 nm of different phospho-forms of C1mC2 normalized to the average signal across each replicate ($n = 4$). Comparisons to the signal of the unphosphorylated C1mC2 were performed using an unpaired t-test and corrected for multiple comparisons with the Benjamini-Hochberg procedure with a false discovery rate of 0.05.

Michaelis-Menten kinetics with substrate competition between sites[32]. For model fitting, we collected additional dephosphorylation time course data on different bis- and trisphosphorylated C1mC2 substrates (Supplementary Fig. 2). Using experimental data from the current study and the phosphorylation data previously published, we then calibrated our model utilizing a repeated and combined global/local parameter search approach followed by selecting the best-performing parameter sets (cf. Methods for details). While many individual turnover and Michaelis-constants were not uniquely identifiable and varied over >2 orders of magnitudes, specificity constants ($k_{cat}/K_m$) typically exhibited much lower variability (Supplementary Fig. 3). To increase the validity of the model predictions, all simulations were performed with multiple independent parameter sets, which allowed us to predict significant effects that were robust despite variability in model parameters.

The developed model reproduces both the PKA phosphorylation and PP1/PP2A dephosphorylation timecourses for substrates with ≥3 phosphate groups. However, we found systematic discrepancies between model output and experimental data for dephosphorylation of bis-phosphorylated forms of C1mC2 (Fig. 2c and Supplementary Figs. 4 and 5). Upon closer inspection of the experimental time course data for dephosphorylation of αβ and αδ (Fig. 2c and Supplementary Fig. 2), it appears as if the accumulation of completely dephosphorylated C1mC2 suddenly slows down in the absence of bis-phosphorylated C1mC2 despite ample supply of mono-phosphorylated substrate for the phosphatases. Because dephosphorylation of mono-phosphorylated C1mC2 appears to directly depend on the presence of bis-phosphorylated C1mC2, we hypothesized that ≥2P-C1mC2 may directly stimulate the dephosphorylation of 1P-C1mC2.

Implementing this assumption into a simple phenomenological model (model 2) improved the quality of the fits as judged by closer agreement with experimental time course data for PP1 and a better score when applying the Akaike information criterion (AIC) for model selection (Supplementary Figs. 6–8). How might such direct dependence of the α-dephosphorylation rate on 2P-C1mC2 work mechanistically?

One potential mechanism is allosteric activation of PP1 and PP2A by phosphorylated (≥2 P) C1mC2 (Fig. 2d, model 3). Allosteric activation of both PP1 and PP2A by peptides or small molecules has been described previously[33–35]. Interestingly, cMyBP-C features an RVXF-motif close to its phosphorylation sites (Supplementary Fig. 9), which might bind to an allosteric site of PP1[32,33]. Alternatively, the dephosphorylation mechanism might be regulated by the conformational states of the substrate. Notably, PKA phosphorylation of cMyBP-C's m-motif has been reported to induce a disorder-to-order transition[36,37]. We, therefore, speculated that dephosphorylation of either the δ- or β-site in bis-phosphorylated C1mC2 may lead to a conformation that is still transiently ordered and allows for rapid dephosphorylation of the α-site before collapsing into a disordered state in which dephosphorylation of the α-site is strongly inhibited (Fig. 2d, model 4). Both mechanistic possibilities considerably improved the model fit to the experimental data in comparison to the initial model and performed better when applying the AIC with, however, similar scores (Fig. 2e and Supplementary Figs. 10 and 11). To decide between the two models, we experimentally tested both hypotheses.

Using pulldown-assays and microscale thermophoresis, we confirmed that C1mC2 can bind PP1 with a high affinity, which is further increased upon thiophosphorylation (Fig. 2f top and Supplementary Fig. 12). However, pre-incubation of PP1 with either thiophosphorylated or Ser-to-Asp substituted C1mC2 to mimic phosphorylation did not increase PP1 activity in both para-Nitrophenylphosphate (pNPP) assays or towards tetrakis-phosphorylated C1mC2 (Fig. 2f, bottom and Supplementary Fig. 13), suggesting that higher phosphorylated forms (≥2 P) of C1mC2 do not allosterically activate PP1.

To experimentally test the structural transition hypothesis (model 4), we sought to obtain experimental evidence using a steady-state approach. If our hypothesis is correct, phosphorylation of C1mC2 at the α-site alone would not be sufficient to push the conformational equilibrium towards an ordered state. To test this prediction, we used the SYPRO-assay: altered fluorescence intensity upon binding of the SYPRO-dye to hydrophobic surface patches is used in thermal denaturation assays to measure protein stability[38]. Since denaturation represents an extreme example of conformational order-to-disorder transition, we reasoned that the signal from the SYPRO assay could be used as a proxy for conformational order. We found that αβ- and αβγ-phosphorylation of C1mC2 led to a significant signal decrease compared to the unphosphorylated protein, suggesting a disorder-to-order transition. In contrast, and in agreement with our prediction, we found that phosphorylation of C1mC2 at the α-site did not decrease the signal, but even led to an increase (Fig. 2g), indicating that phosphorylation at the α-site alone cannot induce a more ordered conformation. Together, these experiments suggest that the structural transition model is more likely to be correct than the allosteric activation model.

## Dose-response of cMyBP-C phosphorylation in the presence of kinases and phosphatases

We implemented the structural transition assumption into our original model and trained the full model with all datasets. As expected, the updated model performs significantly better than the strictly Michaelis-Menten kinetics-based model (Supplementary Figs. 14–16). To complete the model, we added reactions for RSK2 and PKCε and determined their reaction parameters towards C1mC2 experimentally (Supplementary Figs. 17 and 18).

Using this final version of the model, we first simulated the steady-state phosphorylation of cMyBP-C in the presence of kinases and phosphatases. Increasing concentrations of PKA led to a gradual decrease of unphosphorylated cMyBP-C (0P), accompanied by a broad and high peak of 1P, followed by a small peak of 2P and a switch-like transition to 3P cMyBP-C (Fig. 3a). Due to the hierarchical nature of cMyBP-C (de-)phosphorylation by PKA and PP1, these species correspond strictly to α, αβ and αβγ. In addition to parameters, several factors are known to contribute to switch-like responses in multisite phosphorylation systems. If both kinase and phosphatase follow a sequential mechanism, the steepness of the response is generally much higher than if one or both enzymes follow a random mechanism[39,40]. The sequential mechanisms of PKA and PP1/PP2A thus likely support the emergence of the switch-like response in cMyBP-C phosphorylation.

We experimentally tested the predicted dose-response by incubating C1mC2 in the presence of 0.5 µmol/L PP1 with increasing concentrations of PKA and analyzed the steady-state C1mC2 phosphorylation levels using Phostag™-SDS-PAGE. As shown in Fig. 3b, the pattern revealed in the experimental data (i.e., shape of the curves and their relative positions) qualitatively matches the model predictions: A gradual decrease in 0P C1mC2 is followed by broad and small peaks of 1P and 2P, respectively, and a rapid transition to higher phosphorylated species (3P and 4P). In fact, both model and experimental data show a switch-like transition from mainly 1P to mainly 3P species within a narrow range of PKA activity (10–100 nM in experiments) that is well described by the Hill-equation with a coefficient ($n_H$) of about 2 (Fig. 3b, inset and Fig. 3f). A direct comparison of model predictions and experimental data, however, shows that experimental curves are markedly shifted to the right (Supplementary Fig. 19a), which is surprising given the quality of the model fits for both PKA and PP1 data. Further experiments showed that the ATP required for PKA reduces activity of PP1 while the $MnCl_2$ required for PP1 strongly inhibits PKA (Supplementary Fig. 19b, c). Since the effective enzyme activities thus do not match with the activities from the experiments

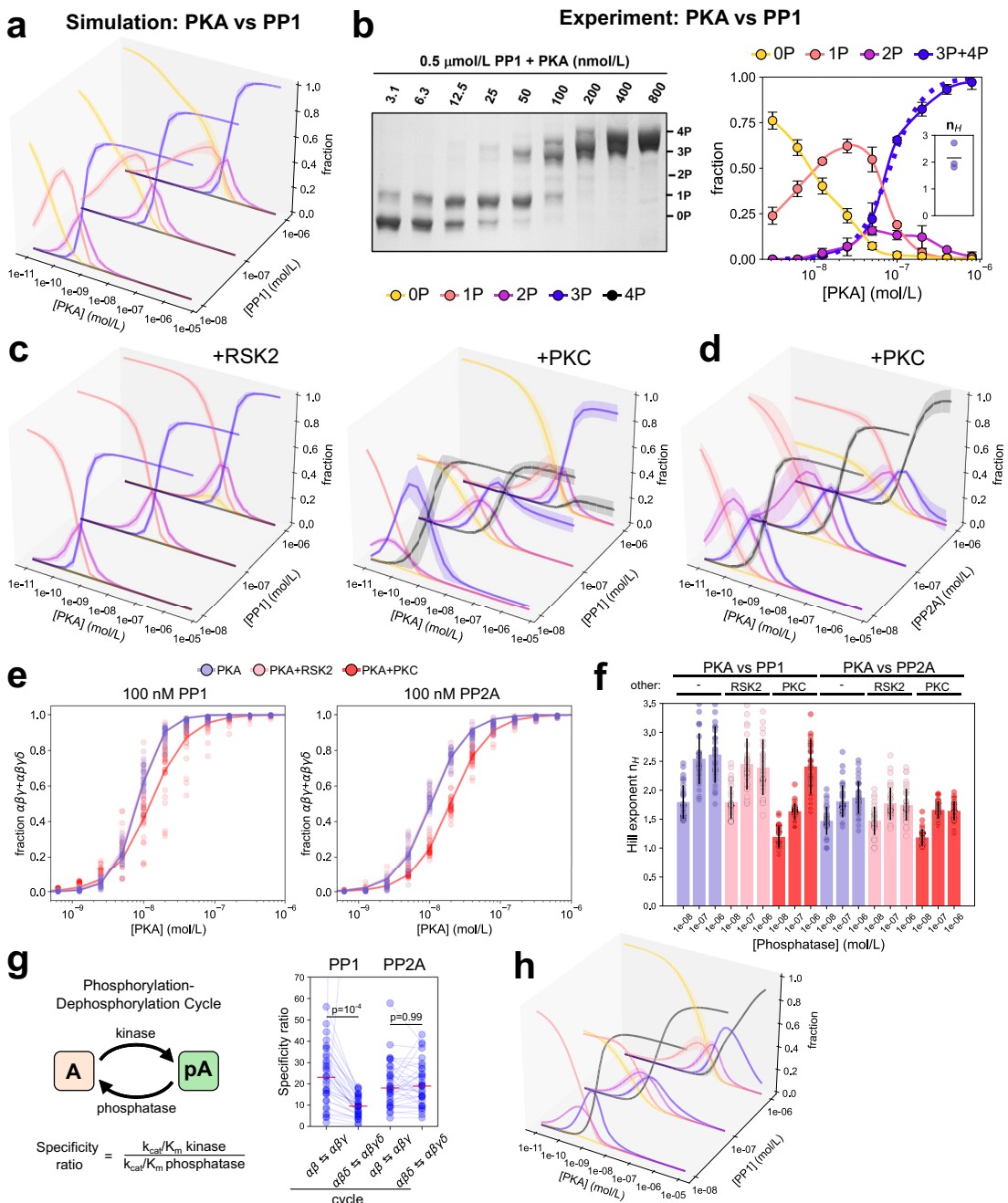

**Fig. 3 | cMyBP-C phosphorylation in presence of kinases and phosphatases.**
**a** Predicted steady-state cMyBP-C phosphorylation response to increasing PKA concentrations and at different concentrations of PP1 (each slice corresponds to a different, fixed phosphatase concentration). 0P: yellow, 1P: orange, 2P: pink, 3P: purple. Mean ± SD of simulations for $n = 35$ parameter sets. **b** left: experimental steady-state C1mC2 phosphorylation levels in the presence of 0.5 µmol/L PP1 and increasing concentrations of PKA analyzed by PhosTag™-SDS-PAGE. Right: Means ± SD for $n = 3$ repeats. The dashed line indicates the average fit of lumped 3P + 4P data using the Hill-equations. Inset: fitted Hill-coefficients. **c** Predicted steady-state cMyBP-C phosphorylation levels determined like in A but in the presence of additional 100 nmol/L RSK2 (left) or PKC (right). **d** Predicted steady-state cMyBP-C phosphorylation levels determined like in A but with PP2A and in the presence of additional 100 nmol/L PKC. Mean ± SD of simulations for $n = 35$

parameter sets. **e** Steady-state dose-responses of lumped cMyBP-C αβγ + αβγδ states to increasing PKA concentrations at a phosphatase, RSK2, or PKC concentration of 100 nmol/L fitted to a Hill-equation to quantify the response. Continuous line shows the fitted average, dots are the individual values for each of the $n = 35$ parameter sets. **f** Extracted Hill coefficient parameters of the predicted PKA-dependent steady-state dose-responses of αβγ + αβγδ cMyBP-C for PP1 and PP2A in the presence or absence of PKC or RSK2. Mean ± SD of simulations for $n = 35$ parameter sets. **g** Specifity ratios of different phosphorylation-dephosphorylation cycles quantifying the influence of the δ-site on γ-site phosphorylation in the presence of PP1 or PP2A. Dots show individual values for each of the $n = 35$ parameter sets. Comparisons were performed with the Mann–Whitney $U$ test. **h** Control simulation with PP1 parameters for reactions αβγδ→αβδ and αβγδ→αβγ exchanged with those of PP2A. Mean ± SD of simulations for $n = 35$ parameter sets.

used for fitting the model, this explains why the experimental dose-response is shifted. The qualitative agreement of the experimental curves, however, still indicates that the model captures important relationships of cMyBP-C regulation by kinases and phosphatases.

Next, we examined the steady-state cMyBP-C phosphorylation responses to PKCε and RSK2 in the presence of PP1, which showed the expected behavior of single-site phosphorylation systems (Supplementary Fig. 20). Due to the reported cross-talk between sites[6], we also

studied the responses to PKA in presence of RSK2 and PKCε (Fig. 3c). In all simulations, the presence of 100 nM RSK2 is sufficient to convert all cMyBP-C into the 1P (α)-state. Further increasing PKA concentration resulted in a gradual decrease of 1P and small 2P (αβ) peak followed by the complete conversion of cMyBP-C into the αβγ state.

The general qualitative features of all responses considered so far are preserved between PP1 and PP2A (Supplementary Figs. 20 and 21). The presence of 100 nM PKCε, however, revealed interesting differences between PP1 and PP2A: for PP1, the endpoints of the 4P- and the biphasic 3P-cMyBP-C curves in response to increasing PKA concentrations strongly depend on the total PP1 concentration (Fig. 3c, right). In contrast, increasing PKA in the presence of PP2A leads to transient intermediate 2P and 3P peaks followed by the near-complete transition to the 4P state for all PP2A concentrations (Fig. 3d). While PKCε dependent cMyBP-C phosphorylation at the δ-site impairs PKA-dependent phosphorylation of the γ-site, the δ-site seems to exert only minor effects on thin filament structure and no effects on thick filament structure and force generation[6]. Phosphorylation of the δ-site therefore likely modifies other cMyBP-C phosphorylation sites without directly influencing sarcomere function itself.

Since the transition to the αβγ and/or αβγδ state in response to PKA often appeared more switch-like for PP1 compared to PP2A, we quantified these properties by fitting the αβγ + αβγδ responses to extract Hill-coefficients that describe the steepness of the response. Interestingly, the addition of PKCε seemed to decrease the steepness of the PKA responses in the presence of PP1, but for PP2A only shifted the response curves to the right (Fig. 3e). Generally, Hill-coefficients were larger for PP1 than for PP2A (Fig. 3f). A full statistical summary of these data is given in Supplementary Table 1.

To better understand why the cMyBP-C phosphorylation response is different in the presence of PP1 and PP2A, especially with respect to 3P and 4P cMyBP-C at high PKA activity, we examined more closely the kinetics of individual reactions. While cMyBP-C phosphorylation at the δ-site impairs PKA-dependent phosphorylation of the γ-site in the absence of phosphatases[6], it is unclear whether this effect still holds in the presence of phosphatases as the δ-site might also impair dephosphorylation. To quantify this, we considered the ratio of the specificity of the kinase for the forward and the specificity of the phosphatase for the backward reaction of a phosphorylation-dephosphoryation cycle as a measure of how much the cycle is biased towards phosphorylation at equal concentrations of kinase and phosphatase. We found that in the presence of PP1, this ratio is higher for the αβ↔αβγ cycle compared to the αβδ↔αβγδ cycle, whereas for PP2A, this ratio did not change (Fig. 3g). This suggests that phosphorylation of the δ-site still impairs phosphorylation at the γ-site in the presence of PP1 but *not* in the presence of PP2A. We tested whether these reactions account for the differences in the dose-response curves by a simulation in which PP1 parameters for the αβγ↔αβγδ and αβδ↔αβγδ cycles were swapped for PP2A parameters and found that the dose-response in presence of PP1 and at high, but not at low PKA concentrations indeed became more similar to the response in presence of PP2A (Fig. 3h). We conclude that the δ-site likely modifies the responsiveness of the remaining sites to other enzymes and that phosphorylation at this site is responsible for differential responses to PKA in the presence of PP1 and PP2A.

Given the independent nature of the δ-site, we re-analyzed the data by lumping together all phosphorylation states, which only differ by the presence or absence of δ-phosphorylation (Supplementary Fig. 22). However, qualitative differences between PP1 and PP2A remained. In particular, at low PP1 concentrations, high levels of αβ + αβδ (>50%) can be reached, whereas at high PP1 concentration, cMyBP-C transitions almost directly from a dominant 0 P+δ state to the αβγ + αβγδ state upon increasing PKA (Supplementary Fig. 22, left). For PP2A, the system transitions invariably from a moderate 0 P+δ state through a small region where αβ + αβδ dominates to αβγ + αβγδ (Supplementary Fig. 22, right).

## Potential effects of HF-associated changes in PKA and phosphatase activities on cMyBP-C phosphorylation

cMyBP-C phosphorylation has been reported to be significantly reduced during HF, potentially due to β-adrenergic receptor desensitization and increases in total phosphatase activity[9,41–46]. To see whether our model, whose parameters were calibrated on in vitro data, is consistent with the cMyBP-C phosphorylation patterns in non-failing or failing donor hearts reported previously[9], we fitted our model to the non-failing heart data set using only the concentrations of PKA, PKCε, RSK2, PP1 and PP2A (restricted to a physiologically plausible range), but not the kinetic constants, as free parameters. The fitted distributions largely capture the experimentally observed relationships between cMyBP-C phosphorylation states in non-failing donor hearts (Fig. 4a left, donor (CL) vs donor (fit)), suggesting that the fitted parameters may reflect the average enzyme activities in non-failing hearts.

During HF, β-adrenergic receptors are strongly desensitized while phosphatase activity in cardiomyocytes is increased[42,43,45,46]. To determine whether these changes can account for the altered cMyBP-C phosphorylation observed in end-stage HF, we used the enzyme concentrations obtained by fitting the model to the data from non-failing heart donors and then decreased PKA concentration and increased total phosphatase concentration. Reliable estimates of PKA activity in the sarcomeric compartment under HF conditions are difficult to obtain (densities of β-adrenergic receptors upstream of PKA under HF can be reduced not at all or by up to ≈75% depending on location in the heart and disease progression[42,47]). As a starting point, we thus assume a ≈50% reduced PKA activity. For phosphatases, we increased PP1 and PP2A by 100% each, following reports that activities and/or expression of PP1 and PP2A are increased ~1.5–2-fold[43,45,46], which together constitutes fourfold decrease in PKA/phosphatase ratio. However, the predicted pattern disagrees with the experimental one and overestimates cMyBP-C phosphorylation (Fig. 4a, HF (predicted) vs HF (CL)).

We thus hypothesized that the PKA/phosphatase ratio may be further decreased and/or that the changes in other enzyme activities might contribute to HF-related changes in cMyBP-C phosphorylation. Therefore, we fitted our model directly to the failing heart data while keeping only a two-fold increase in total phosphatase activity as a constraint (Fig. 4a, (HF (CL) vs HF (fit)). When comparing the estimated enzyme concentrations for the non-failing and failing donor hearts, we indeed found further differences in addition to the expected decrease in [PKA] such as a significant increase in [PKCε] by a factor of about four (Fig. 4b). Moreover, the fitted enzyme concentrations amount to a ten-fold decrease in the PKA/phosphatase ratio (Fig. 4b, right), explaining why a four-fold reduction in PKA/phosphatase ratio was insufficient to predict cMyBP-C patterns in failing hearts. We also observed a small increase in RSK2 concentrations during HF, which, however, was not significant, indicating that RSK2 is not required to explain these data. Strikingly, the two-fold increase in total phosphatase concentration was almost exclusively a result of increased [PP1], whereas [PP2A] did not change much. For non-failing donor hearts, in contrast, the fitted PP1 concentration was very low compared to the PP2A concentration. This suggests that the phosphatase concentrations in the sarcomeric compartment may differ markedly from the approximately 2-fold increased total phosphatase concentrations during HF. To gain further insights into the relevance of these predictions, we fixed the PP1/PP2A ratio to PP1/PPases$_{tot}$ = 1 for donor hearts and to PP2A/PPases$_{tot}$ = 1 for failing hearts. Interestingly, we found the quality of the fits to be notably reduced (Supplementary Fig. 23), suggesting that the ratio between these phosphatases is an important determinant of the cMyBP-C phosphorylation state.

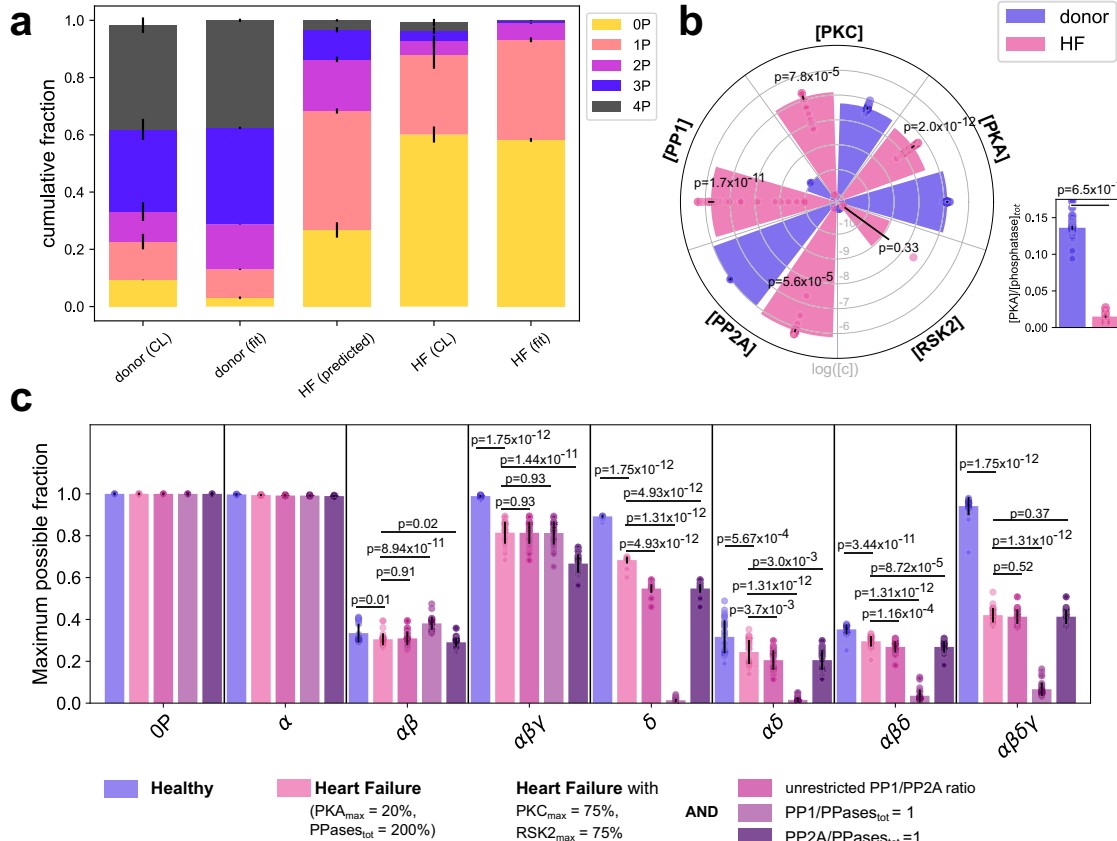

**Fig. 4 | cMyBP-C phosphorylation states during heart failure (HF). a** The model was tested for consistency with experimental data on cMyBP-C basal phosphorylation states in hearts from healthy donors or HF patients reported in Copeland et al. 2010 (CL) by fitting the model using only the enzyme concentrations as free parameters (fit). Additionally, the distribution of cMyBP-C phosphorylation states during HF was predicted by starting with the enzyme concentrations fitted to the donor (CL) data followed by a decrease in [PKA] by 50% and a 2-fold increase of PP1 and PP2A concentrations consistent with previous reports on β-adrenergic receptor downregulation and phosphatase activity during HF. **b** Comparison of the fitted enzyme concentrations underlying donor (fit) and HF (fit) data from **a** and calculated PKA/phosphatase_tot ratios. **c** Maximally achievable fraction for each cMyBP-C

phosphorylation state under various conditions. For each phosphorylation state, algorithmic optimization was used to find the enzyme vector [PKA, PKC, RSK2, PP1, PP2A] (within a physiological range) that maximizes the phosphorylation state under consideration. To probe the effect of perturbing other enzymes during HF, PKC and RSK2 concentrations were further restricted and PP1/PP2A ratio was either allowed to vary freely (crimson), or phosphatases were fixed to PP2A/PPase_tot = 1 (purple) or PP1/PPase_tot = 1 (dark purple). Each fitting or optimization run has been performed with all of the $n = 35$ parameter sets. Values represent mean ± SEM and comparisons in (**b**, **c**) were done via Mann–Whitney $U$ test corrected for multiple comparisons with the Benjamini–Hochberg procedure with a false discovery rate of 0.05.

These findings indicate that the remodeling process of the signaling networks controling cMyBP-C phosphorylation during HF could be more complex than previously assumed. Specifically, we wondered whether changes in PKCε activity or in the PP1/PP2A ratio might even have beneficial effects by allowing cMyBP-C to access phosphorylation states that otherwise might be precluded by β-adrenergic receptor desensitization or increased total phosphatase activity. To address this issue and gain more functional insights, we rephrased this question as an optimization problem and asked: to what degree can individual phosphorylation states be maximized and what are the required enzyme concentrations?

Guided by the concentrations resulting from fitting the model to donor heart data, we identified the maximally achievable fractions (MAFs) for each of the eight cMyBP-C phosphorylation states and their corresponding enzyme vectors under various conditions (Fig. 4c and Supplementary Figs. 24–28). The purpose of calculating the MAFs is to evaluate under which restrictions (in terms of enzyme concentrations) and for which sites the phosphorylation states are reduced and under which other conditions these reductions can still be compensated. Thereby, this analysis may help to disentangle which of the observed changes in Fig. 4b may be causative and which ones may be compensatory.

Similar to our dose-response data, we found that the MAFs for intermediate states (αβ, αδ, αβδ) are generally lower than for other states. Under HF conditions, the MAFs of many states were unaffected, but moderate to strong reductions were found for αδ, αβγ, αβδ and αβγδ (Fig. 4c, blue vs red). Since these correspond to 2P, 3P, and 4P species, this observation is consistent with the strongly reduced amount of 3P and 4P cMyBP-C in the failing donor hearts, indicating that these reductions cannot be sufficiently compensated during HF. To better understand the contribution of individual enzymes that might be altered by HF-related remodeling, we repeated the analysis by further restricting the concentrations of RSK2 and PKCε and/or by clamping the relative amounts of PP1 and PP2A (without changing total phosphatase concentrations). Interestingly, while reducing RSK2 and PKCε showed little effect (except on δ), a significant reduction of αβγ could be seen when fixing PP2A/PPase_tot to 1, and almost complete abolishment of δ, αδ, αβδ, and αβγδ when fixing PP1/PPase_tot to 1.

So far, experiments and analyses have been performed with cMyBP-C substrate concentrations much higher than enzyme concentrations. In vivo, however, substrate and enzyme concentrations may be comparable, violating the assumptions of the Michaelis-Menten approach. We thus repeated the analyses with a total-quasi-steady-state rate law valid at lower substrate concentrations[48,49] but

found no major qualitative differences, except that the change in PP1/PP2A ratio was even more pronounced due to a predicted decrease of [PP2A] during HF (Supplementary Figs. 29 and 30).

Together, these findings indicate that changes in the PP1/PP2A ratio may indeed compensate for some of the HF-associated changes irrespective of total phosphatase or substrate concentration.

## Discussion

Previous studies reported that both PP1 and PP2A are able to dephosphorylate cMyBP-C in vivo[27–29]. Although we showed in the present study that the cMyBP-C dephosphorylation timecourses appear largely similar in vitro, we found significant differences in the specificities for individual sites that resulted in pronounced differences between PP1 and PP2A in the PKA responses, particularly in the presence of PKC (Fig. 3c, d). Another important finding of both of our simulations and experiments is the apparent cooperativity (switch-like response) of the cMyBP-C phosphorylation state (Fig. 3), suggesting that the system can be sensitive to small changes in the PKA/phosphatase activity (reflected by the steepness of the [PKA]-[3P/4P] relationship with Hill-exponents ≥2). As shown in Fig. 3a, changing active PKA concentration by a factor of two leads to a re-distribution of the cMyBP-C phosphorylation state between a mostly tetrakis/trisphosphorylated species (~70% of total) and an almost entirely mono- and unphosphorylated species. This suggests that at some threshold sympathetic tone, β-adrenergic signaling likely leads to robust transition into a regime with high cMyBP-C phosphorylation and its associated functional consequences[17]. Perhaps such a switch-like transition is one of the main functions of cMyBP-C multisite phosphorylation/dephosphorylation. Similar switch-like responses stemming from multisite (de-)phosphorylation have been described recently for phospholamban[50]. Since the Hill-exponent does not increase with lower phosphatase concentration (Fig. 3f), the effect is likely supported by the sequential cMyBP-C (de-)phosphorylation mechanism rather than zero-order ultrasensitivity[39,51], although changes of the Hill-exponent in response to PKCε and differences between PP1 and PP2A suggest this property can be tuned by reaction parameters, too.

Together, these data suggest that PP1 and PP2A may play non-redundant roles in the regulation of cMyBP-C function. The differences between both phosphatases may not only be necessary to differentially regulate contractile function in response to various receptor signals, but changes in the activity of either enzyme could also be a factor in the pathophysiology of HF. We thus confirmed the validity of our model for cMyBP-C phosphorylation in basal and disease conditions using experimental data from human heart donors and HF patients[9]. Our analyses suggest that the observed cMyBP-C hypo-phosphorylation during HF might not be solely explained by decreasing PKA and increasing phosphatase activity. Indeed, the biggest contributor to the observed hypo-phosphorylation in our simulations is a shift in the ratio of PP1 to PP2A activity, and changing this ratio could potentially increase cMyBP-C phosphorylation in the HF setting. Additionally, our model also predicted PKCε activity acting on cMyBP-C to be significantly elevated. Although the precise role of α-adrenergic signaling (activating PKC) in heart muscle function has remained elusive, both in vivo studies in animal models and isolated intact muscle preparations showed a robust positive inotropic response after α1-adrenergic receptor stimulation[52,53]. Our previous study[6] and our current findings, however, highlight a cross-talk between α- and β-adrenergic stimulation on the level of the contractile myofilaments. PKCε phosphorylation of cMyBP-C not only increases the $EC_{50}$ for phosphorylation of cMyBP-C by PKA but also reduces its sensitivity (Fig. 3). Thus, α-adrenergic signaling via PKCε might allow a more graded transition of cMyBP-C phosphorylation states, which allows access to intermediate phosphorylation states that are not accessible during exclusive PKA/β-adrenergic stimulation (e.g., 2 P states). In good agreement, the bis-phosphorylated state of cMyBP-

C is underrepresented under basal conditions in both human and rodent hearts[9], suggesting that this state might be more accessible during disease conditions. However, PKD too has been reported to phosphorylate the δ-site of cMyBP-C and could thus play a similar role alternatively or in addition[30]. Yet, while our model predicts that activity changes in multiple enzymes are required to explain the observed cMyBP-C phosphorylation pattern during HF, the reported changes in the literature (particularly for phosphatases PP1 and PP2A) are somewhat inconsistent, perhaps reflecting differences in experimental models, etiology or disease progression[43,45,46,54,55].

Concerning the interpretation of our results pertaining to the human heart data, there are two important limitations to consider. Firstly, these predictions rely on the assumption that the optimized in vitro conditions used for the enzymes studied here and the models based on this data can tell us something relevant about the situation in vivo. However, as shown in Supplementary Fig. 19, enzymatic assays can be quite sensitive to a variety of factors. The results on human heart data and the predictions about changes in enzyme activities during HF are thus best understood as consistency checks and testable hypotheses about qualitative changes in the signaling activities acting on cMyBP-C in the failing heart rather than precise quantitative predictions. A second, closely related limitation is that the current model is incomplete with respect to components further upstream in the cardiac signaling network (or CamKII, PKD, and GSK3, for that matter), some of which likely have important implications for cMyBP-C phosphorylation. In contrast to the in vitro situation of our study, many enzyme activities at the sarcomere are regulated in a highly dynamic manner due to regulation by various subunits in vivo. Substrate dephosphorylation by PP1, for example, is subject to regulation by feedforward loops involving PKA, PKC, and inhibitor-1[50,56], whereas sarcomeric PP2A activity is regulated by B56-subunits, e.g., via changed localization after β-adrenergic stimulation[28,57,58] and Pak1[29], which itself lies downstream of multiple receptor tyrosine kinases. Translocation to the myofilaments in response to β-adrenergic stimulation has also been reported for PKCε[59]. How these dynamic signaling processes impact cMyBP-C phosphorylation via differential regulation of PP1 and PP2A (along with kinases) and which extracellular signals are responsible for this is poorly understood. Both limitations thus require further experimental work based on cell biological or in vivo experiments.

Here, we have provided a detailed and site-specific biochemical characterization of cMyBP-C dephosphorylation by PP1 and PP2A and developed an integrated model of cMyBP-C phosphorylation as a function of multiple kinases and phosphatases. Our study not only highlights the importance of phosphatases for regulating cMyBP-C phosphorylation, but also provides a useful framework for integrating data on cMyBP-C phosphorylation from cells or animals in future studies. Moreover, modulating cMyBP-C phosphorylation is increasingly considered a potentially promising therapeutic strategy[60–63]. An integrated and validated model of how cMyBP-C is regulated by both phosphatases and kinases could be used to design therapeutic strategies to optimize cMyBP-C phosphorylation in heart disease by modulating multiple rather than single enzymatic activities. Such a strategy could not only prove more effective but may require lower doses than single-target therapies. Since cMyBP-C is a hub for pathogenic mutations, it further would be interesting to study the effect of relevant mutations on (de)phoshorylation and how physiological cMyBP-C phosphorylation patterns may be restored.

## Methods
### Recombinant proteins
The catalytic subunit $PP1_\beta^{6-327}$ was cloned into a pET15b vector and validated via sequencing by Bio Basic (USA). For purification of PP1, 50 μL SoluBL21™ Chemically Competent E. coli (AMSBIO, UK) cells were transformed with pET15b-$PP1_\beta^{6-327}$ and a 100 mL preculture was grown in LB-medium overnight at 37 °C in a shaking incubator at

200 rpm. On the following morning $2 \times 1$ L autoinduction medium[64] supplemented with 1 mM MnCl$_2$ were inoculated with 50 mL of the preculture each and further grown at 30 °C until the evening before the temperature was lowered to 20 °C and cultures were left to grow for ~72 hrs before harvesting. After harvesting, cells were resuspended in lysis buffer (50 mM TRIS pH 8, 700 mM NaCl, 1 mM MnCl$_2$, 1 mM DTT, 25 mM Imidazole, cOmplete™ EDTA-free Protease Inhibitor Cocktail (Roche, cat. no. 11873580001)) and lyzed via 2× freeze/thaw cycles followed by addition of lysozyme, the addition of Triton X-100 ad 0.2%, and sonication. After clearing the lysate from insoluble material via centrifugation and filtration, PP1 was purified via a HisTrap™ FF column (GE Healthcare, cat. no. 29048609) followed by size exclusion chromatography (Superdex™ 75 pg, GE Healthcare, cat. no. 28-9893-34) into a final buffer of 50 mM TRIS pH 8, 700 mM NaCl, 1 mM MnCl$_2$, 1 mM DTT. The protein was concentrated to a final concentration of ≈0.2 mg/ml (determined by absorbance at 280 nm), aliquoted, snapfrozen in liquid nitrogen, and stored at −70 °C.

The phosphorylated cMyBP-C fragments encompassing domains C1, m-motif, and C2 (C1mC2) were prepared using PKA (Calbiochem, cat. no. 539576), PKCε (Sigma, cat. no. SRP5067) and RSK2 (Millipore, cat. no. 14-480) as described previously[1,6]. The homogeneity of phospho-species was determined by electron-spray ionization mass spectrometry. Phosphorylated C1mC2 was gel-filtered into assay buffer (in mmol/L: 20 MOPS pH 7, 100 KCl, 1 MgCl$_2$, 1 MnCl$_2$, 2 DTT, 0.05% (v/v) Tween-20) using NAP5 columns (Life Technologies) according to manufacturer's instructions and protein concentrations adjusted to 20 μmol/L.

Protein phosphatase 2 A (PP2A) was purchased from Cayman Chemical (cat. no. 10011237), aliquoted, and stored at −70 °C until further use. To avoid activity loss, enzymes were not refrozen after thawing of aliquots.

## Pulldown experiments

Pulldown experiments for testing whether the catalytic domain of PP1 and cMyBP-C C1mC2 interact with each other were performed as follows: first, 50 μL Ni-NTA resin (Qiagen, cat. no. 30410) was washed three times in assay buffer (PBS pH 7.5, 500 mM NaCl, 0.01% tween-20, 40 mM Imidazole, 0.5 mM MnCl$_2$, 2 mM DTT) in 1.5 ml tubes before 100 μL of 1 μM Alexa647-labeled (Thermo Fischer, cat. no. A37573) BSA (negative control for bait) or C1mC2 were added to the resin. Next, 75 μL of 5 μM His$_6$-PP1 was added to resin with A647-C1mC2 or A647-BSA (negative control). In another negative control, 75 μL 1× PBS was added to resin with A647-C1mC2. Tubes were incubated for 45 min at RT in a tube shaker at 1000 rpm before a sample of the supernatant was taken and mixed with 5× SDS sample buffer as a reference for the unbound fraction (SDS sample buffer:sample = 1:5). The supernatant was removed, the resin was washed three times in 750 μL assay buffer and the resin was resuspended in 50 μL 1× SDS sample buffer and boiled for 5 min at 100 °C. For SDS-PAGE analysis, 12.5 μL sample/lane were run on a gel. After running the gel, the A647 signal was recorded, and the gel was stained with Coomassie solution to obtain the total protein signal (Protein Ark, cat. no. GEN-QC-STAIN-1L).

## Microscale thermophoresis

For quantifying the interaction strength between PP1 and cMyBP-C, binding affinities were determined via Microscale Thermophoresis on a Monolith NT.115 Instrument (NanoTemper Technologies, Germany). Briefly, serially diluted cMyBP-C C1mC2 was mixed 1:1 with 100 nM Alexa647-PP1 in MST buffer (50 mmol/L Tris-HCl pH 8, 500 mmol/L NaCl, 1 mmol/L MnCl$_2$, 2 mmol/L DTT, 0.05% (v/v) tween-20, 1 mmol/L MnCl$_2$ and 1 mmol/L DTT were added just before the experiment) and incubated at RT for 5 min before samples were loaded with premium capillaries (NanoTemper Technologies, cat. no. MO-K025). Data were recorded sequentially at 40%, 90%, and 95% MST power and 25% laser power. The best data quality was obtained at 90%

MST power which was thus used for quantifying the binding affinities using the MO. Affinity Analysis software (NanoTemper Technologies, Germany).

## pNPP assay

The p-nitrophenyl phosphate (pNPP) assay was used to test for potential allosteric activation of PP1 by cMyBP-C C1mC2. Briefly, 100 nmol/L PP1 were preincubated with no additional factors or with 2 μmol/L unphosphorylated or thiophosphorylated cMyBP-C C1mC2 in assay buffer (50 mmol/L Tris-HCl pH 8, 500 mmol/L NaCl, 1 mmol/L MnCl$_2$, 2 mmol/L DTT, 0.05% (v/v) tween-20, 1 mmol/L MnCl$_2$ and 1 mmol/L DTT were added just before the experiment) for 5 min at 30 °C. The preincubated PP1 sample was then mixed 1:1 with a freshly made solution of pNPP (Sigma Aldrich, cat. no. 20-106) in serially diluted assay buffer to reach a final concentration of 50 nmol/L PP1 and ~1–100 mmol/L pNPP in 96-well microplates (Greiner bio-one, cat. no. 675076). pNPP turnover was recorded by measuring absorbance at 405 nm over time in a CLARIOstar microplate reader (BMG LABTECH, Germany). The initial rates obtained from the recorded traces were fitted to the Michaelis-Menten equation to estimate $V_{max}$ and $K_m$ values.

## (De)phosphorylation assays

Dephosphorylation reaction timecourses were started by adding 0.1 μmol/L PP1 or PP2A to 20 μmol/L 3P/4P-C1mC2 in assay buffer (in mmol/L: 20 MOPS pH 7, 100 KCl, 1 MgCl$_2$, 1 MnCl$_2$, 2 DTT, 0.05% (v/v) Tween-20), and aliquots quenched at the indicated time points with 5x SDS-PAGE loading buffer and snap frozen in liquid nitrogen. Michaelis-Menten kinetics for dephosphorylation reactions were performed by incubating varying concentrations of site-specifically phosphorylated C1mC2 in assay buffer (in mmol/L: 20 MOPS pH 7, 100 KCl, 1 MgCl$_2$, 1 MnCl$_2$, 2 DTT, 0.05% (v/v) Tween-20) with 0.1 μmol/L PP1 or PP2A and quenching the reactions early enough for a sufficient approximation of the initial rates $v_0$ (we used 2 min for pS308-C1mC2 and 10 min for pS288-C1mC2).

Michaelis-Menten kinetics for PKCε and RSK2 were performed similarly by incubating different substrate concentrations (C1mC2-0P) with 1 U/mL kinase, and the reactions quenched after 5 mins with SDS-PAGE loading buffer and heat denaturation, and phospho-species separated by Phostag™-SDS-PAGE as described below.

Dose-response experiments were performed by incubating 20 μmol/L 0P-C1mC2 and 0.5 μmol/L PP1 with different PKA concentrations in an assay buffer containing 200 μmol/L ATP. Reactions were quenched after 60 min incubation at 30 °C (to allow the system to reach a steady state) with SDS-PAGE loading buffer and heat denaturation, and phospho-species separated by Phostag™-SDS-PAGE as described below.

In each case, samples were denatured at 100 °C for 3 mins, and equal amounts of protein were loaded onto a 12% (v/v) acrylamide SDS-PAGE gel containing 50 μmol/L Phostag™-reagent and 100 μmol/L MnCl$_2$. Gels were run at 160 V for 90 mins and protein bands visualized using Coomassie staining. Concentrations of different C1mC2 phospho-species in each sample were calculated from the ratio of the signal of the band corresponding to the respective phosphorylated or unphosphorylated species and the total signal of a lane multiplied by the known total concentration of C1mC2. In some timecourses (e.g., Fig. 1b), another band between 2P and 3P bands is visible. Although the nature of this phospho-species is currently unknown, it has been shown that Phostag-SDS-PAGE not only separates phosphorylated proteins according to the number of phosphorylated residues but also according to their respective position in the peptide[65]. It follows that this minor species might correspond to a 2P or a 3P species that is less abundant than other 2P or 3P species, e.g., it could be a pS288, pS308 bis-phosphorylated C1mC2 fragment. However, as it was not possible to decide between these options and

since this band was far less abundant than other bands, we did not take this band into account except for quantifying the total protein amount per lane.

## Western blotting

SDS-PAGE samples were run on 4–20% (v/v) acrylamide gradient gels (BIORAD) and transferred onto nitrocellulose membranes using a Trans-Blot SD Semi-Dry Electrophoretic Transfer Cell (BioRad). Primary incubation took place overnight at 4 °C (anti-pSer279; 1:5000 dilution; anti-pSer288, 1:1000 dilution; anti-pS308, 1:10000 dilution). Antibodies were produced by ProSci Inc. as described previously[14]. Briefly, antibodies were raised in rabbits against phospho-peptides using the murine cMyBP-C sequence (AFRRT(pS)LAGAG for anti-pS279, GAGRRT(pS)DSHEDA for anti-pS288, LKKRD(pS)FRRDS for anti-pS308) and purified via two columns (first column with non-phosphorylated peptide to remove antibodies recognizing non-phosphorylated peptides, second column with phosphopeptide to specifically enrich the antibodies against the phosphorylated target peptide). Specificity of generated antibodies was confimed by ProSci Inc. via ELISA (see Source Data File) using the corresponding phospho-peptides and by Western-blot (Supplementary Fig. 31). Secondary antibody incubation was done at room temperature for 1 h (HRP-conjugated goat anti-rabbit IgG, 1:2000 dilution, BioRad, 403005). Membranes were washed with Tris-buffered saline containing 0.05% (v/v) Tween-20, emersed in ECL reagent (BIORAD), and blots developed on a BIORAD imager.

## SYPRO fluorescence assay

SYPRO orange was purchased from Life Technologies as a 10000-fold stock solution in DMSO. 20 µmol/L unphosphorylated and phosphorylated C1mC2 fragments were incubated with 20-fold SYPRO Orange for 10 min at 25 °C in black 96-well plates (Greiner), and SYPRO Orange fluorescence was measured using appropriate excitation and emission filter settings using a ClarioStar plate reader (BMG Labtech).

## Identification of phosphorylation states in 2P cMyBP-C during dephosphorylation using mass spectrometry

To identify the sites that are still phosphorylated in the bis-phosphorylated cMyBP-C C1mC2 fragment during the dephosphorylation transient starting from the almost phosphorylated C1mC2 fragment, the dephosphorylation reaction was stopped 3–5 min after addition of PP1 or PP2A by addition of SDS sample buffer and 5 min boiling at 100 °C. Samples were separated by PhosTag-SDS-PAGE, and 2 P bands ($n = 1$) were cut out and sent to the Metabolomics and Proteomics Laboratory of the Bioscience Technology Facility of the University of York for identification of phosphorylation sites, which carried out the following procedures:

In-gel tryptic digestion was performed after reduction with dithioerythritol and S-carbamidomethylation with iodoacetamide. Gel pieces were washed two times with aqueous 50% (v:v) acetonitrile containing 25 mM ammonium bicarbonate, then once with acetonitrile and dried in a vacuum concentrator for 20 min. Sequencing-grade, modified porcine trypsin (Promega) was dissolved in 50 mM acetic acid, then diluted fivefold with 25 mM ammonium bicarbonate to give a final trypsin concentration of 0.02 µg/µL. Gel pieces were rehydrated by adding 25 µL of trypsin solution, and after 10 min enough 25 mM ammonium bicarbonate solution was added to cover the gel pieces. Digests were incubated overnight at 37 °C. Peptides were extracted by washing three times with aqueous 50% (v:v) acetonitrile containing 0.1% (v:v) trifluoroacetic acid, before drying in a vacuum concentrator.

Dry peptides were brought up in aqueous 1 M glycolic acid containing 80% acetonitrile and 5% trifluoroacetic acid for phosphopeptide enrichment using MagReSyn® TiO2 nano-spheres (MR-TID02) following the manufacturer's standard protocol (https://resynbio.com/wp-content/uploads/2022/09/IFU_TiO2.pdf). Bound peptides

were washed with 80% acetonitrile containing 1% trifluoroacetic acid, then 10% acetonitrile with 0.2% trifluoroacetic acid before eluting with aqueous 1% NH4OH. Enriched phospho-peptides were acidified with formic acid before drying in a vacuum concentrator.

Peptides were resuspended in aqueous 0.1% trifluoroacetic acid (v/v) then loaded onto an mClass nanoflow UPLC system (Waters) equipped with a nanoEaze M/Z Symmetry 100 Å C18, 5 µm trap column (180 µm × 20 mm, Waters) and a PepMap, 2 µm, 100 Å, C 18 EasyNano nanocapillary column (75 µm × 500 mm, Thermo Fischer). The trap wash solvent was aqueous 0.05% (v:v) trifluoroacetic acid and the trapping flow rate was 15 µL/min. The trap was washed for 5 min before switching flow to the capillary column. Separation used gradient elution of two solvents: solvent A, aqueous 0.1% (v:v) formic acid; solvent B, acetonitrile containing 0.1% (v:v) formic acid. The flow rate for the capillary column was 300 nL/min, and the column temperature was 40 °C. The linear multi-step gradient profile was: 3–10% B over 7 mins, 10–35% B over 30 mins, 35–99% B over 5 mins, and then proceeded to wash with 99% solvent B for 4 min. The column was returned to initial conditions and re-equilibrated for 15 min before subsequent injections.

The nanoLC system was interfaced with an Orbitrap Fusion Tribrid mass spectrometer (Thermo Fischer) with an EasyNano ionization source (Thermo Fischer). Positive ESI-MS and MS 2 spectra were acquired using Xcalibur software (version 4.0, Thermo). Instrument source settings were: ion spray voltage, 1900–2100 V; sweep gas, 0 Arb; ion transfer tube temperature; 275 °C. MS1 spectra were acquired in the Orbitrap with: 120,000 resolution, scan range: m/z 375–1500; AGC target, 4e5; max fill time, 100 ms. Data-dependant acquisition was performed in top speed mode using a 1 s cycle, selecting the most intense precursors with charge states >1. Easy-IC was used for internal calibration. Dynamic exclusion was performed for 50 s post precursor selection, and a minimum threshold for fragmentation was set at 5e3. MS2 spectra were acquired in the linear ion trap with: scan rate, turbo; quadrupole isolation, 1.6 m/z; activation type, HCD; activation energy: 32%; AGC target, 5e3; first mass, 110 m/z; max fill time, 100 ms. Acquisitions were arranged by Xcalibur to inject ions for all available parallelizable time.

**PTMs were searched using PEAKS software with the following search parameters.** PEAKS Version: PEAKS Studio 10.6 build 20201015; Search Engine Name: PEAKS; Parent Mass Error Tolerance: 3.0 ppm; Fragment Mass Error Tolerance: 0.5 Da; Precursor Mass Search Type: monoisotopic; Enzyme: Trypsin; Max Missed Cleavages: 2; Digest Mode: Specific Fixed; Modifications: Carbamidomethylation: 57.02; Variable Modifications: Oxidation (M): 15.99; Phosphorylation (STY): 79.97; Max Variable PTM Per Peptide: 2; Database: protein1; Taxon: All; Contaminant Database: cRAP Searched Entry: 118; FDR Estimation; Enabled Merge Options: no merge Precursor; Options: corrected Charge; Options: no correction Filter; Charge: 2–10 Process: true; Associate chimera: yes; Ion Source: ESI(nano-spray); Fragmentation Mode: high energy CID (y and b ions); MS Scan Mode: FT-ICR/Orbitrap; MS/MS Scan Mode: Linear Ion Trap.

## Mathematical models

All models have been formulated using ordinary differential equations that describe the rates of phosphorylation and dephosphorylation of cMyBP-C forms. Generally, enzymatic reactions were described by Michaelis−Menten type rate laws that account for the possibility of substrate competition[32] between different phospho-forms: $v_i = \frac{k_{cat,i}\, e s_i}{K_i(1 + \kappa_E - \frac{s_i}{K_i}) + s_i}$, where $e$ denotes the concentration of Enzyme E, $s_i$ the concentration of the ith substrate of E, $k_{cat,i}$ and $K_i$ are the respective turnover and Michaelis-constants and $\kappa_E = \sum \frac{s_j}{K_j}$ (sum over all

substrates $S_j$ of E). For the extended model versions allowing a better description of 2P-cMyBP-C dephosphorylation, the rates were modified as follows:

For the phenomenological model, the rate of α-dephosphorylation was multiplied with $(1 + f_{act} \, h)$, where the parameter $f_{act} \geq 0$ denotes a maximum activation factor and $h = \frac{r_{2P/3P}}{K_{act} + r_{2P/3P}}$ is a hyperbolic function of $r_{2P/3P}$, the relative amount of bis- and trisphosphorylated cMyBP-C, with a half-saturation constant $K_{act}$.

For the allosteric activation model, we first derived a general steady-state rate law for allosteric activation for catalysis of a substrate $S_j$ (single-site) and multiple non-allosterically competing substrates. Consider the following reaction scheme below where A is an allosteric activator for the catalysis of substrate $S_j$ to product $P_j$ in the presence of multiple competing substrates whose rate of catalysis is not affected by A:

We assume that the total amount of enzyme $e_T$ is conserved and given by the relation $e_T = e + e_A + c_j + c_A + \sum_{i=1}^{n} c_i$. The rate at which product $P_j$ is formed is given by $v = k_j \cdot e_j + k_A \cdot e_A$, therefore yielding equation 1: $\frac{v}{e_T} = \frac{k_j \cdot e_j + k_A \cdot e_A}{e + e_A + c_j + c_A + \sum_{i=1}^{n} c_i}$. We further assume the rates of complex formation are much faster than catalytic turnover, allowing us to use the rapid equilibrium approximation as outlined in ref. 66 for equilibria $K_j$, $K_A$, $\lambda K_j$, $\lambda K_A$ and $K_1, \ldots, K_n$, where $\lambda$ is a dimensionless scaling parameter to describe the preferential formation of the allosterically activated complex $c_A$. Solving the equilibria for the enzyme or complex species and substituting the respective terms in equation 1 yields the final form of the rate law:

$$v = \frac{k_j \cdot e_T \cdot s_j + k_A \cdot \frac{e_T \cdot s_j \cdot a}{\lambda K_A}}{K_j + \frac{e_T \cdot a}{K_A} + \frac{s_j \cdot a}{\lambda K_A} + K_j(\sum_{i=1}^{n} c_i) + s_j}.$$ We used this rate law to describe the dephosphorylation of α, assuming for simplicity that only 2P-cMyBP-C species serve as allosteric activators, i.e., $A = \alpha\beta + \alpha\delta$.

For the structural transition model, we assumed that dephosphorylation of the β-phosphate from αβ or the δ-phosphate from αδ leads to the production of α', a (transiently) ordered conformation of α which is a better substrate for phosphatases PP1 and PP2A than α. Isomerization between α' and α was assumed to follow first-order rates $v_{iso,F} = k_{iso,F} \bullet \alpha'$ and $v_{iso,R} = k_{iso,R} \bullet \alpha$, respectively.

To study the system response at low substrate concentrations (i.e., of similar order of magnitude to the enzyme concentrations) at which the assumptions of Michaelis–Menten type rate laws are violated, we also used a total-quasi-steady-state version of the model following the approach from[48,49]. Following this approach, the tQSSA rate law for an irreversible distributive multisite phosphorylation system such as the one considered here can be given by $v_j = \frac{k_j \cdot e_T \cdot s_j}{(K_j + e_T)(1 + \sum_{i=1}^{n} \frac{s_i}{K_i + e_T})}$, where $j$ is the index of the $j$th reaction step in the sequence.

Further details and ODEs for all models can be found in the SI Appendix.

For simulation, all models were implemented in Python and integrated with a custom implementation of the fourth-order Runge–Kutta integrator as outlined in[67] with a step-size of 1 s. If necessary, step-size

was reduced to 0.1 or 0.01 s or an LSODA integrator from the Python package scipy (v1.9.3) was used instead.

## Model fitting, selection, and optimization

For parameter estimation, models were implemented in COPASI v4.35[68] and parameter estimation was performed with Python using the package basico (v0.4, https://github.com/copasi/basico)[69]: For each model, 50–100 independent parameter sets were generated using a global search procedure (Genetic algorithm, generations: 250, population size: 25) and subsequently refined using a local search (Hooke–Jeeves-algorithm, iteration limit 50, tolerance $10^{-5}$, rho 0.2). Generally, weights were assigned by standard deviation and normalized by experiment. In case of obvious fitting artifacts (unrealistic transients resulting from limited temporal resolution in the experimental data), custom weights were assigned manually. To filter out parameter sets with remaining artifacts, we calculated the sum of squared errors between simulated data and the Akima-interpolation of the experimental data. All parameter sets leading to an error that lies above a predefined cutoff for any of the experiments were filtered out (some cutoffs were set by trial and error but generally mean + 0.5 standard deviations worked for most conditions).

For model selection, we utilized the Akaike Information Criterion[70]. The AIC score for each model and parameter set was calculated as described in ref. 71. A model was judged to be superior to another one if it had a statistically significantly lower AIC score.

For fitting the model to the data from Copeland et al.[9] each of the 35 parameter sets resulting from fitting the model to our in vitro data was sequentially applied to a Copasi implementation of the final model which was then used for parameter estimation with enzyme concentrations of PKA, PKC, RSK2, PP1 and total phosphatase concentration as free parameters (Hooke-Jeeves-algorithm, iteration limit 50, tolerance $10^{-5}$, rho 0.2). Based on the previous literature[72] and the kinetic parameters in the present study, enzyme concentrations in the high nanomolar to low micromolar range seemed plausible to us. Specifically, parameter search was conducted within the following concentration ranges: PKA $\in [5 \times 10^{-10}, 5 \times 10^{-7}]$, PKC $\in [0, 2 \times 10^{-7}]$, RSK2 $\in [0, 2 \times 10^{-7}]$, PP1$\in [10^{-10}, 10^{-6}]$ and total phosphatases $\in [10^{-10}, 10^{-6}]$ (all in mol/L). PP2A concentration was calculated by total phosphatase concentration minus PP1 concentration, thus the fraction that PP1 or PP2A contributes to total phosphatase concentration was allowed to vary freely. For fitting the HF data, the total phosphatase concentration was fixed to twice the total phosphatase concentration obtained by fitting the model to the donor heart data. For control experiments shown in Supplementary Fig. 22, phosphatases were set to either PP1/PPases$_{tot}$ = 1 or PP1/PPases$_{tot}$ = 1.

Similarly, for optimizing individual phosphorylation states, each of the 35 parameter sets resulting from fitting the model to our in vitro data was sequentially applied to a COPASI implementation of the final model, which was then used for maximizing a chosen cMyBP-C phosphorylation state with enzyme concentrations of PKA, PKC, RSK2, PP1 and total phosphatase concentration as free parameters (Hooke-Jeeves-algorithm, iteration limit 50, tolerance $10^{-5}$, rho 0.2). Permissible enzyme ranges were guided by the resulting concentrations after fitting the model to the data from ref. 9. More specifically, we assumed that PKA and total phosphatase concentrations resulting from fitting the model to the cMyBP-C phosphorylation data from non-failing hearts represent basal activities in the absence of any β-adrenergic or other modulatory stimuli. While we kept total phosphatase concentrations constant, we reasoned that β-adrenergic stimulation might increase PKA concentration by at least 10x or that cholinergic stimulation might further depress basal PKA activity. PKC, RSK2, and PP1/PP2A ratio was subject to the same constraints as before. For optimizing individual cMyBP-C phosphorylation states under conditions resembling a healthy heart, the ranges were thus PKA $\in [0, 10\text{xPKA}_{donor(fit)}] \approx [0, 1.36 \times 10^{-6}]$, PKC $\in [0, 2 \times 10^{-7}]$, RSK2 $\in [0, 2 \times 10^{-7}]$, PP1 $\in [0, \text{PPases}_{tot,donor(fit)}]$, total

phosphatases = $PPases_{tot,donor(fit)} \approx 10^{-6}$ (all in mol/L). For optimizing individual cMyBP-C phosphorylation states under conditions resembling a failing heart, total phosphatase concentration was increased twofold in line with previous reports that both PP1 and PP2A increase by up to two-fold[43,45,46], while we assumed that the reduction in $PKA_{HF(fit)}$ compared to $PKA_{donor(fit)}$ at least partly is a result of β-adrenergic receptor desensitization and therefore also applies to conditions of high β-adrenergic stimulation. We thus choose $PKA \in [0, 10xPKA_{HF(fit)}] \approx [0, 2.89 \times 10^{-7}]$, $PKC \in [0, 2 \times 10^{-7}]$, $RSK2 \in [0, 2 \times 10^{-7}]$, $PP1 \in [0, PPases_{tot,HF(fit)}]$, total phosphatases = $PPases_{tot,HF(fit)} \approx 2 \times 10^{-6}$ (all in mol/L). To further probe the functional effect of individual enzymes, we also tested the following conditions (all concentrations in mol/L):

- HF with PKC/RSK2 restriction:
  $PKA \in [0, 10xPKA_{HF(fit)}] \approx [0, 2.89 \times 10^{-7}]$, $PKC \in [0, 1.5 \times 10^{-7}]$, $RSK2 \in [0, 1.5 \times 10^{-7}]$, $PP1 \in [0, PPases_{tot,HF(fit)}]$, total phosphatases = $PPases_{tot,HF(fit)} \approx 2 \times 10^{-6}$

- HF with PKC/RSK2 restriction and clamped PP1:
  $PKA \in [0, 10xPKA_{HF(fit)}] \approx [0, 2.89 \times 10^{-7}]$, $PKC \in [0, 1.5 \times 10^{-7}]$, $RSK2 \in [0, 1.5 \times 10^{-7}]$, $PP1 = PPases_{tot,HF(fit)}$, total phosphatases = $PPases_{tot,HF(fit)} \approx 2 \times 10^{-6}$

- HF with PKC/RSK2 restriction and clamped PP2A:
  $PKA \in [0, 10xPKA_{HF(fit)}] \approx [0, 2.89 \times 10^{-7}]$, $PKC \in [0, 1.5 \times 10^{-7}]$, $RSK2 \in [0, 1.5 \times 10^{-7}]$, $PP1 = 0$, total phosphatases = $PPases_{tot,HF(fit)} \approx 2 \times 10^{-6}$

### Statistical analysis
Data were analyzed in GraphPad Prism or Python using the packages numpy (v1.21.5), scipy (stats module, v1.9.3), and statsmodels (v0.13.2). Statistical comparisons between two normal distributions have been performed using a two-sided t-test and Welch's *t* test for unequal variances. Distributions from simulations often appeared non-normal, which was confirmed using the Shapiro-Wilk test (scipy stats). For comparisons between two non-normally distributed variables, the Mann–Whitney test was used (scipy stats). Comparisons between multiple groups were performed using the Kruskal–Wallis test (scipy stats) or by a 1-way ANOVA (in case of AIC values on the $log_2$-normalized distributions), followed by Tukey's multiple comparison tests where relevant. In the case of multiple comparisons, *p* values were corrected using the Benjamini/Hochberg procedure with a false discovery rate of 0.05 (statsmodels). For any test, a *p* value below 0.05 was considered to indicate a statistically significant difference.

### Reporting summary
Further information on research design is available in the Nature Portfolio Reporting Summary linked to this article.

## Data availability
The mass spectrometry data generated in this study have been deposited in the massIVE database under accession code MSV000091995. Source data are provided with this paper.

## Code availability
All codes generated in this study have been deposited in the GitHub repository [https://github.com/KochLabCode/cMyBP-C-phosphorylation][73].

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

## Acknowledgements
We would like to thank Aneta Koseska for helpful feedback on earlier versions of the manuscript, Frank Bergmann for help with the basico package, and Shodhan Rao for help with the derivation of the tQSSA rate law. LC-MS/MS analysis was performed by the University of York's Bioscience Technology Facility, Metabolomics and Proteomics Laboratory, Department of Biology, University of York, UK, using instrumentation within the York Center of Excellence in Mass Spectrometry (CoEMS). The York CoEMS was created thanks to a major capital investment through Science City York, supported by Yorkshire Forward with funds from the Northern Way Initiative, and subsequent support from EPSRC (EP/K039660/1; EP/M028127/1). We acknowledge the British Heart Foundation (grant nr: FS/16/3/31887, T.K.) and the European Molecular Biology Organization (grant nr: ALTF 310-2021, D.K.) for financial support.

## Author contributions
T.K.: conceptualization, investigation, writing—review & editing. S.P.: resources, writing—review & editing. K.S.C.: resources, writing—review & editing. A.W-H.: resources, writing—review & editing. D.K.: conceptualization, supervision, formal analysis, investigation, writing—original draft.

## Funding

## Competing interests
The authors declare no competing interests.
