## [Peer Review File · Nature Communications]

Reviewers' Comments:

Reviewer #1:

Remarks to the Author:

Cardiac myosin binding protein-C (cMyBP-C) plays a vital role in regulating myofilament and cardiac function. Its study holds clinical relevance due to its strong genetic association with hypertrophic cardiomyopathy. In this study, the authors focus on a crucial aspect of cMyBP-C biology: the process of cMyBP-C dephosphorylation. Understanding this process is essential, as the dephosphorylation of cMyBP-C occurs in various forms of cardiac stress and contributes to cardiac dysfunction. While extensive research has been conducted on cMyBP-C phosphorylation, little attention has been given to the mechanisms driving this pathological dephosphorylation. The authors employ a comprehensive approach, combining biochemical experiments and computational modeling, to unravel the hierarchical mechanism underlying cMyBP-C dephosphorylation. They also apply their computational model to analyze cMyBP-C phosphorylation in human heart failure. Overall, this outstanding contribution sheds light on an underappreciated area of the myofilament field, and the computational modeling of competitive kinase-phosphatase systems holds broad interest. One general comment is that the results sections can be dense and difficult to follow, particularly due to insufficient explanations of certain experiments. Simplifying and making these results more accessible would greatly benefit a generalist audience. For example, the Michaelis-Menten and pNPP assays require clarification, while the Sypro Ruby assay is well described.

Major:

Figure 1c: It is recommended to include the phosphorylation status of other C1mC2 phosphosites in this experiment to confirm their rapid dephosphorylation. Phospho-specific antibodies exist and should be utilized.

Figure 1d: This experiment is challenging to follow, and the text should provide more detailed information on the methodology and readouts. It should be clarified whether C1mC2 was phosphorylated by RSKIII or PKC, and additional details such as phosphatase concentration, timings, and assay readouts need to be included. Is the ratio of phosphorylated to non-phosphorylated C1mC2 quantified in the Michaelis-Menten plot?

Figure 1e: In the main text, a more comprehensive methodology should be provided for the mass spectrometry data, including the phosphosite KAIGSpSG. The current presentation of the data suggests a low peptide read-depth, and the generation of proteolytic peptides may bias the detection of S279 and S288 compared to S308 and S313. A simplified approach to presenting this data could involve comparing the phosphospecies ratio of tetrakis- and bis-phosphorylated C1mC2. This approach would provide quantitative analysis, and the inclusion of tetrakis-phosphorylated C1mC2 would address concerns regarding potential biases in phosphopeptide detection.

Methods: A detailed explanation of Michaelis-Menten kinetics and mass spectrometry methods should be included. Additionally, the supplementary data file "Phos_ID_D428" is difficult to locate.

Minor:

Introduction paragraph 2: In the final sentence, the use of "consistently" implies more than one study and should therefore also cite Tong CW, et al. *Circ Res.* 2008 (PMID: 18802026).

Figure 1b, PP1: It appears that there may be an intermediate phosphospecies at 5 and 10 minutes between the 3P and 2P bands. Have the authors investigated the possibility of an additional phosphosite within their phosphorylated C1mC2? The mass spectrometry data seems to support this hypothesis and would be of interest.

Reviewer #2:

Remarks to the Author:

The manuscript by T. Kampourakis and colleagues 'Cardiac myosin binding protein-C phosphorylation ...' combines biochemical and mathematical modeling approaches to study the detailed regulation of cardiac myosin binding protein-C (cMyBP-C) by phosphorylation. The biological and biomedical relevance of the study is high because of cMyBP-C's central role in regulating heart muscle contractility via multiple input signals affecting the protein's phosphorylation status. This regulation, however, is only partially understood, and the original approach taken by the authors is to focus on the role of phosphatases, to complement the predominant prior work on kinases. The main claimed results are:

- (i) site-specific and hierarchical control of phosphorylation by the two phosphatases considered (PP1 and PP2A);
- (ii) sequential dephosphorylation in opposite direction of the main kinase (PKA);
- (iii) development of an integrated mathematical model for site-specific regulation that proposes new mechanisms such as allosteric activation or ordered-disordered protein transition;
- (iv) functional differences between PP1 and PP2A action in the presence of protein kinase C (PKC); and
- (v) consistency of the model with (human) patient data for heart failure – upon model re-parametrization – that suggests non-redundant physiological roles of PP1 and PP2A (as well as additional regulatory mechanisms).

Overall, the manuscript is well-structured and presented, providing all required data for reproduction. The experimental part of the study is based on careful (in vitro) experiments with isolated proteins that provide strong evidence for claims (i) and (ii) as well as partly (iii). In addition, the experiments are used to specifically test and validate competing hypotheses in conjunction with the mathematical model (variants), for example, on protein conformational changes. While this provides an excellent basis for model development, and thereby for the analysis of patient data, the mathematical model and its corresponding inferences for claims (iv) and (v) appear less rigorous as detailed below, such that the inferred physiological role of phosphatase regulation of cMyBP-C does not appear sufficiently supported.

Major comments:

(i) Model structure: To capture multisite phosphorylation of cMyBP-C by multiple kinases and phosphatases, Michaelis-Menten type kinetics with substrate competition are used. This approach has two main limitations that are likely to affect mechanistic interpretations of the model and in particular the inference from human data (claims iv and v): (i) these kinetics do not consider enzyme competition for the same substrate explicitly (akin to competition of multiple substrates for the same enzyme) and (ii) more importantly, while the central assumption of Michaelis-Menten kinetics of excess substrate over enzyme may hold for the in vitro experiments, it is unlikely to be correct for the in vivo situation. For the reasons detailed in e.g. Salazar & Hofer, FEBS J. 276 (2009) the model-based analysis should proceed from elementary reaction kinetics, with a concomitant increase in the number of parameters. The latter challenge can be reduced by justified approximations (e.g. quasi steady-state or common time-scales for reaction types).

(ii) Statistical model validation: The use of AIC values demonstrates superior description of the data for the ultimate model 4, but it does not answer the question if any of the models sufficiently captures the data. The analysis should therefore be augmented by a test for model fit given the data and their uncertainty, such as χ^2 tests, also to support the frequent use of 'significant' in the text (e.g., related to Fig. 2c). In terms of statistics, two additional aspects should be considered: (i) tests should be performed on unnormalized data or with appropriate variance propagation (e.g., the data in Fig. 2g suggests no variance for the OP form), and (ii) model selection via AIC relies on the maximum likelihood estimate for a given set of optimally parametrized models and a given data set; variance in model statistics as in Fig. 2e results from

sub-optimal model parametrizations and should not be used for model selection, but model probabilities can be computed directly from the maximum likelihood estimates.

(iii) Simulation scenarios for heart failure: As demonstrated by the preceding simulations (e.g., Fig. 3) predicted cMyBP-C phosphorylation patterns depend on the quantity of PKA. PKA activity reduction by 50% is one change to the 'normal' model to predict cMyBP-C phosphorylation in heart failure, but no justification for the value is given. For the two-fold increase in phosphatase, the references do not seem to support the chosen value: ref. 40 does not contain data on PP1/PP2A, and ref. 41 demonstrates only two-fold activity increase of PP1. If experimental justifications of these values are impossible, simulation studies with different (plausible) PKA activity levels should be conducted to achieve robust conclusions, e.g., the influence of PP1 vs PP2A.

Minor comments:

(i) p.2, 'regulat[i]on'

(ii) p.3., second para: Please explain why no other kinases than PKA, RSK2, and PKC (e.g., PKD) were included.

(iii) p.3, 3rd para and other instances: Please use 'significant' (e.g. for slower dephosphorylation) only when supported by a statistical test.

(iv) Fig. 1d: For later arguments on differences between PP1 and PP2A, one could test for statistical differences of K_M/k_{cat} values.

(v) p.6, 'Allosteric activation of both ...' – would such activation by specific phosphorylation sites of cMyBP-C be sterically possible?

(vi) Fig. 3a,b and elsewhere: To compare model predictions and experimental data, please combine them in the same plot (not only a Hill function fit as in Fig. 3b).

(vii) p.12 and Fig. 4c: Please explain in more detail why MAFs are a suitable characterization for the different conditions, and how these are consistent with the patient data (e.g. 'but moderate to strong reductions ...' which are 2P,3P,and 4P).

(viii) p.13 'the effect is likely supported by the sequential ...': Cooperative binding is another potential mechanisms (see Salazar and Hofer above) that could be checked in the parametrized model.

(ix) p.17, 'custom weights': These weights need justification, and a (statistically) cleaner solution would be to omit them.

(x) Fig. S21: It is unclear, which predictions the experimental data are compared to (and how the statement in the main text is supported).

Reviewer #3:

Remarks to the Author:

In this paper, Kampourakis, Ponnam, and Koch describes a mathematical model for the phosphorylation status of cMyBP-C taking both kinases and phosphatases into account, especially PP1 and PP2A. It is important to take all different kinases and phosphatases into account when modelling the reaction kinetics, and the authors do a good job including many of them – but, as they mention themselves, further components also play a role in the cMyBP-C phosphorylation equilibrium. In my opinion, the authors do a good job adjusting/fitting the kinetic model. However,

the data the model is based on could be more convincing. The paper is very well written. As I am not an expert in cardiac biology or mathematical modelling, my comments mainly concern the mass spectrometric data.

General points

- To my knowledge nature journals require four replicates for statistical analysis. In most cases the authors report n=1-3.
- The multi-phosphorylated peptide is essential for the story. The authors reference a previous study for the generation of this, but including a short description on how it was made (and how the phosphorylation site specificity checked) would be valuable.
- The concept of making kinetic models taking several kinases and phosphatases into account is not trivial. Having excellent data to base the models on is a necessity. The data presented in this paper could be more convincing. One idea could be to analyze the phosphorylated peptide using mass spectrometry, making sure all four phosphosites are identified as a start. The peptide could then be incubated with the different phosphatases, or a combination, and samples taken for mass spectrometric analysis at different time points – and maybe normalized to the intensity/abundance of the individual sites based on the control sample (with all four sites). And, all of this performed in quadruplicates of course to ensure statistical significance.
- While I appreciate that the authors developed the first integrated model of cMyBP-C phosphorylation as a function of multiple kinases and phosphatases, it would be a good addition to the story, if they could speculate on how this could be used or would be beneficial in the future.

Comments to specific pages

- Page 3: The authors write: "Physiologically, the interaction of the cMyBP-C with both filament systems is controlled by phosphorylation at several sites in the cardiac-specific m-motif by physiologically relevant protein kinases". Maybe change to "Physiologically, the interaction of the cMyBP-C with both filament systems is controlled by phosphorylation at several sites in the cardiac-specific m-motif by protein kinases".
- Page 3: For the sentence above, maybe the references should be added before the reference to the figure?
- Page 4: The authors use a 50% mixture of the peptide containing 3 or 4 phosphorylation sites. Why is this necessary? Is it in order to be able to distinguish them using PhosTag?
- Page 4: The authors claim that because they only identify pS279 and pS288 using mass spectrometry, pS308 and pS313 are likely dephosphorylated first. I am not convinced by this. Absence of evidence is not equal evidence of absence. The experiment lacks a proper control and replicates (see points regarding mass spectrometry data below).
- Page 5, figure 1c: The authors claim that because the pS288 signal is relatively stable, and the signal correlates with that for the mono-phosphorylated species, that pS288 likely accounts for the slow dephosphorylation phase. This is a rather strong conclusion, and would benefit from more evidence. The authors could for example consider including a positive control by including a condition where a phosphatase capable of removing phosphorylation on pS288 was included (if one exists?). This would also serve as a control for the antibody.
- Page 5, figure 1d: This figure is not clear to me. To me it looks like there is more of the dephosphorylated form when treating with lower concentrations of the phosphatase? I also do not see a clear difference between the two blots/substrates.
- Page 6: It would be great if the authors could add more experimental evidence to support their proposed reaction mechanism for cMyBP-C dephosphorylation.
- Page 6, figure S1: It is great that the authors include additional data for the various variants of phosphorylation. However, it would be great if the authors could specify how they generated the unique phospho-versions, and check that it is actually only this species present.
- Page 11, figure 3: I understand that it can be difficult to include many things in one graph, but the 3D graphs are not clear to me.

Points regarding the mass spectrometry data

- The authors include two mass spectrometric RAW files. To account for biological and technical variations, replicates are needed.
- It was not trivial downloading the RAW files from the MassIVE repertoire. In my experience, ProteomeXchange is preferred for uploading MS data.
- The methods available through MassIVE are missing some essential details. It states that "enriched phosphopeptides were loaded...", but how were the phosphopeptides enriched? Details regarding the gradient on the LC and the settings on the MS are also missing.
- The authors/mass spec facility search the data using Peaks. To my knowledge, this software is not the best for localizing PTMs. Search settings are also not described.
- I tried researching the two RAW files using MaxQuant (free software) with cMyBP as a FASTA file, and phosphorylation on serine, threonine, and tyrosine as variable modifications. From this search, pS279 is the most intense/abundant site, and pS288 the second most intense/abundant. A small fraction of the intensity was observed on pS290, and even less on pS312. As MaxQuant reports the localization probability (how likely is it that the modification is actually residing on this reported amino acid residue), it should be noted, that the localization probability of pS288 is only 0.63, whereas class I phosphosites have a localization probability above 0.75. This is not necessarily a problem, if the spectrum looks good (I did not check this).
- In order to conclude anything from mass spectrometric data, a control sample with all four phosphorylation sites identified (and localized) is needed. The two sites not found by mass spectrometry (pS308 and pS313) reside in a sequence surrounded by lysines and arginines, making the trypsin-digested peptides likely too short to identify as peptides often need to be minimum 7 amino acids long. This could potentially be overcome by using different proteases. But in order to use mass spectrometry to proof which phosphosites are present, the phosphosites identified under phosphatase treated conditions should be compared directly to phosphosites in an untreated sample.

General remarks.

We would like to thank the editor and the reviewers for their constructive criticism of our manuscript. We have made substantial revisions based on their suggestions that we believe have further improved our manuscript. The most important changes include the addition of further experiments using site-specific antibodies to address comments and mechanistic concerns from reviewer #1 and #3 as well as tQSSA variant of the model to address the question of the system behaviour at low substrate concentrations raised by reviewer #2. We also explained our reasons whenever we decided not to follow a suggestion. Please find our responses to the individual points raised by the reviewers on a point-by-point basis below.

REVIEWER COMMENTS

Reviewer #1 (Remarks to the Author):

Cardiac myosin binding protein-C (cMyBP-C) plays a vital role in regulating myofilament and cardiac function. Its study holds clinical relevance due to its strong genetic association with hypertrophic cardiomyopathy. In this study, the authors focus on a crucial aspect of cMyBP-C biology: the process of cMyBP-C dephosphorylation. Understanding this process is essential, as the dephosphorylation of cMyBP-C occurs in various forms of cardiac stress and contributes to cardiac dysfunction. While extensive research has been conducted on cMyBP-C phosphorylation, little attention has been given to the mechanisms driving this pathological dephosphorylation. The authors employ a comprehensive approach, combining biochemical experiments and computational modeling, to unravel the hierarchical mechanism underlying cMyBP-C dephosphorylation. They also apply their computational model to analyze cMyBP-C phosphorylation in human heart failure. Overall, this outstanding contribution sheds light on an underappreciated area of the myofilament field, and the computational modeling of competitive kinase-phosphatase systems holds broad interest. One general comment is that the results sections can be dense and difficult to follow, particularly due to insufficient explanations of certain experiments. Simplifying and making these results more accessible would greatly benefit a generalist audience. For example, the Michaelis-Menten and pNPP assays require clarification, while the Sypro Ruby assay is well described.

Major:

Figure 1c: It is recommended to include the phosphorylation status of other C1mC2 phosphosites in this experiment to confirm their rapid dephosphorylation. Phospho-specific antibodies exist and should be utilized.

Authors' response:

We thank the reviewer for this suggestion. We have prepared phospho-specific antibodies for the three major phosphorylation sites and used them in Western-blot experiments (new Figure 1c). The results of the Western-blot experiments are in excellent agreement with the mass spectrometry and biochemical data, supporting a hierarchical de-phosphorylation mechanism.

Figure 1d: This experiment is challenging to follow, and the text should provide more detailed information on the methodology and readouts. It should be clarified whether C1mC2 was phosphorylated by RSKIII or PKC, and additional details such as phosphatase concentration, timings, and assay readouts need to be included.

Authors' response: *Many thanks for pointing out the lack of clarity. We have provided the requested details both in the figure legend and the method section.*

Is the ratio of phosphorylated to non-phosphorylated C1mC2 quantified in the Michaelis-Menten plot?

Authors' response: *The plot shows initial rates v_0 (in the previous version we erroneously wrote k_{obs} instead of v_0 , which we have corrected now) divided by enzyme concentration (the limit of which approaches k_{cat} as the enzyme concentration increases). To calculate the initial rates we quantified the product after quenching the reaction using the ratio of the*

phosphorylated to non-phosphorylated C1mC2 and the known total concentration of C1mC2. These details are now also described in the methods.

Figure 1e: In the main text, a more comprehensive methodology should be provided for the mass spectrometry data, including the phosphosite KAIGSpSG. The current presentation of the data suggests a low peptide read-depth, and the generation of proteolytic peptides may bias the detection of S279 and S288 compared to S308 and S313. A simplified approach to presenting this data could involve comparing the phosphospecies ratio of tetrakis- and bis-phosphorylated C1mC2. This approach would provide quantitative analysis, and the inclusion of tetrakis-phosphorylated C1mC2 would address concerns regarding potential biases in phosphopeptide detection.

Authors' response:

We have added additional mass spectrometry data to the manuscript confirming the position of the phosphorylation sites in PKA tris-, bis and mono-phosphorylated C1mC2, and RSK2-mono-phosphorylated C1mC2, all of which were also previously described in Ponnam et al. 2019.

The PKCepsilon site was previously identified in rodent cardiac myosin binding protein-C, please see Xiao et al., 2007, Biochemistry).

Additionally, the data are in excellent agreement with the Western-blot data that was generated in response to point #1 raised by the referee.

We have edited the manuscript accordingly.

Methods: A detailed explanation of Michaelis-Menten kinetics and mass spectrometry methods should be included. Additionally, the supplementary data file "Phos_ID_D428" is difficult to locate.

Authors' reply: *As mentioned in the response to the comment on Figure 1d, we have provided the requested details on the Michaelis-Menten kinetics in the method section. We also now provide a detailed description of the mass spectrometry procedures. We trust the publishers that any supplementary material will be easy to find alongside the manuscript.*

Minor:

Introduction paragraph 2: In the final sentence, the use of "consistently" implies more than one study and should therefore also cite Tong CW, et al. Circ Res. 2008 (PMID: 18802026).

Authors' reply: *We added the suggested reference.*

Figure 1b, PP1: It appears that there may be an intermediate phosphospecies at 5 and 10 minutes between the 3P and 2P bands. Have the authors investigated the possibility of an additional phosphosite within their phosphorylated C1mC2? The mass spectrometry data seems to support this hypothesis and would be of interest.

Authors' reply: *The reviewer raises a very interesting point. Although the nature of this phospho-species is currently unknown, it has been shown that Phostag-SDS-PAGE not only separates phosphorylated proteins according to the number of phosphorylated residues but also according to their respective position in the peptide (Oerd and Loog, Intrinsically*

Disordered Proteins, 2020, pp779-792). It follows that this minor species might correspond to a 2P or a 3P species that is less abundant than other 2P or 3P species, e.g. it could be a pS288, pS308 bisphosphorylated C1mC2 fragment. However, as it was not possible to decide between these options and since this band was far less abundant than other bands, we did not take this band into account except for quantifying the total protein amount per lane. We have included this explanation into the method section on phostag-SDS-PAGE.

Reviewer #2 (Remarks to the Author):

The manuscript by T. Kampourakis and colleagues ‘Cardiac myosin binding protein-C phosphorylation ...’ combines biochemical and mathematical modeling approaches to study the detailed regulation of cardiac myosin binding protein-C (cMyBP-C) by phosphorylation. The biological and biomedical relevance of the study is high because of cMyBP-C’s central role in regulating heart muscle contractility via multiple input signals affecting the protein’s phosphorylation status. This regulation, however, is only partially understood, and the original approach taken by the authors is to focus on the role of phosphatases, to complement the predominant prior work on kinases. The main claimed results are:

- (i) site-specific and hierarchical control of phosphorylation by the two phosphatases considered (PP1 and PP2A);
- (ii) sequential dephosphorylation in opposite direction of the main kinase (PKA);
- (iii) development of an integrated mathematical model for site-specific regulation that proposes new mechanisms such as allosteric activation or ordered-disordered protein transition;
- (iv) functional differences between PP1 and PP2A action in the presence of protein kinase C (PKC); and
- (v) consistency of the model with (human) patient data for heart failure – upon model re-parametrization – that suggests non-redundant physiological roles of PP1 and PP2A (as well as additional regulatory mechanisms).

Overall, the manuscript is well-structured and presented, providing all required data for reproduction. The experimental part of the study is based on careful (in vitro) experiments with isolated proteins that provide strong evidence for claims (i) and (ii) as well as partly (iii). In addition, the experiments are used to specifically test and validate competing hypotheses in conjunction with the mathematical model (variants), for example, on protein conformational changes. While this provides an excellent basis for model development, and thereby for the analysis of patient data, the mathematical model and its corresponding inferences for claims (iv) and (v) appear less rigorous as detailed below, such that the inferred physiological role of phosphatase regulation of cMyBP-C does not appear sufficiently supported.

Major comments:

(i) Model structure: To capture multisite phosphorylation of cMyBP-C by multiple kinases and phosphatases, Michaelis-Menten type kinetics with substrate competition are used. This approach has two main limitations that are likely to affect mechanistic interpretations of the model and in particular the inference from human data (claims iv and v): (i) these kinetics do not consider enzyme competition for the same substrate explicitly (akin to competition of multiple substrates for the same enzyme) and (ii) more importantly, while the central assumption of Michaelis-Menten kinetics of excess substrate over enzyme may hold for the in vitro experiments, it is unlikely to be correct for the in vivo situation. For the reasons detailed in e.g. Salazar & Hoefler, FEBS J. 276 (2009) the model-based analysis should proceed from elementary reaction kinetics, with a concomitant increase in the number of parameters. The latter challenge can be reduced by justified approximations (e.g. quasi steady-state or common time-scales for reaction types).

Authors’ response: *We agree with the referee on these points. We found the differential-algebraic equation based approach by Salazar & Hoefler impractical for our study – especially as we rely on Copasi for many of our analyses, which does not support differential-algebraic-*

equation systems. Instead, we opted for the proposed (total) quasi-steady-steady approach and repeated the analyses shown in Figure 4 (that are partly relying on *in vivo* data) at 10-100x lower substrate concentrations. Quantitatively, the PP1/PP2A ratio at low substrate concentrations appears even more relevant for fitting the HF data (cf. Figure S29-30) as PP2A decreases at lower substrate concentrations in the HF conditions, leading to a even more altered PP1/PP2A ratio under HF conditions. The overall fit to human data and PKA/phosphatase ratio are almost unchanged. Qualitatively, the analyses thereby show no major differences to the Michaelis-Menten version of the model. Thus, it appears the qualitative model behaviour is generally determined more by the ratios between the substrate and the different enzymes rather than the absolute concentrations.

(ii) Statistical model validation: The use of AIC values demonstrates superior description of the data for the ultimate model 4, but it does not answer the question if any of the models sufficiently captures the data. The analysis should therefore be augmented by a test for model fit given the data and their uncertainty, such as χ^2 tests, also to support the frequent use of 'significant' in the text (e.g., related to Fig. 2c). In terms of statistics, two additional aspects should be considered: (i) tests should be performed on unnormalized data or with appropriate variance propagation (e.g., the data in Fig. 2g suggests no variance for the OP form), and (ii) model selection via AIC relies on the maximum likelihood estimate for a given set of optimally parametrized models and a given data set; variance in model statistics as in Fig. 2e results from sub-optimal model parametrizations and should not be used for model selection, but model probabilities can be computed directly from the maximum likelihood estimates.

Authors' response: We thank the reviewer for these comments. Regarding the lack of variance for the OP data in Figure 2g: small pipetting errors during the dye preparation lead to rather big differences in the absolute signal strength across experiments, making normalization necessary for comparisons based on multiple replicates. Instead of normalizing to the OP sample, we now normalize all samples of a replicate by the average signal across samples of a replicate, thereby making sure information about the variability is not lost due to the normalization.

Regarding the variance of the model parametrization and model selection we take a different stance. We think the variability of the model parametrization stems not from sub-optimal parameter estimation procedures, but instead that the experimental data used for model fitting, even though very rich compared to other studies in the field, is not sufficient to uniquely constrain all of the >50 parameters. Essentially, the variability observed is likely a case of 'universal parameter sloppiness' (<https://journals.plos.org/ploscompbiol/article?id=10.1371/journal.pcbi.0030189>). However, as the authors of the cited paper argued, this is not necessarily a problem for the predictive capability of the model. Note also that the specificity constants (often taken as the fundamental parameter of a reaction as they can be considered the second order rate constants determining the velocity of the reaction at low substrate concentrations; cf. A. Cornish-Bowden (2012) *Fundamentals of Enzyme Kinetics*, p.35-38) of most reactions are much more constrained than the corresponding k_{cat} and K_m values. We further do not think more statistical analyses (e.g. χ^2 tests) will necessarily provide an answer to the question if the models sufficiently describe the data and prefer to leave it to the readers to judge the quality of the fits shown in Figure 2 and in the supplementary figures. While the models and fits are not perfect, the experiments and models in this study, to our knowledge, provide the best quantitative *in vitro* analysis of cMyBP-C phospho-regulation to date. In our view, the most important question is whether the model is good enough for the purpose of the study,

which was to investigate if and how the phosphorylation state of cMyBP-C depends in a non-trivial way not only on the kinases, but on the joint activities of both kinases and phosphatases and their interactions at different phosphorylation sites. Given the mechanistic insights and testable (as well as partly tested) predictions provided, we think the answer to this question is positive.

(iii) Simulation scenarios for heart failure: As demonstrated by the preceding simulations (e.g., Fig. 3) predicted cMyBP-C phosphorylation patterns depend on the quantity of PKA. PKA activity reduction by 50% is one change to the 'normal' model to predict cMyBP-C phosphorylation in heart failure, but no justification for the value is given. For the two-fold increase in phosphatase, the references do not seem to support the chosen value: ref. 40 does not contain data on PP1/PP2A, and ref. 41 demonstrates only two-fold activity increase of PP1. If experimental justifications of these values are impossible, simulation studies with different (plausible) PKA activity levels should be conducted to achieve robust conclusions, e.g., the influence of PP1 vs PP2A.

Authors' response: *We thank the reviewer for pointing out the lack of clarity. The data from Neumann et al. 1997 (ref. 41) indeed reports a 2x increase in PP1 activity, not overall phosphatase activity. We have fixed this mistake, repeated the analyses shown in Figure 4a,b and refer to further literature which justifies the chosen values.*

Generally, reliable estimates of PKA concentration in the sarcomeric compartment under conditions of heart failure are difficult to obtain. In Bristow et al. 1996 (ref 40 in previous version), β -adrenergic receptor density was decreased by approximately by 50%. Although consistent with other studies, the reduction can also be less severe or more pronounced, depending e.g. on disease progression and location in the ventricle ([https://doi.org/10.1016/0735-1097\(89\)90181-2](https://doi.org/10.1016/0735-1097(89)90181-2)). We thus opted for 50% and assumed a 50% reduction of PKA activity as a rough estimate as it is unknown how the reduction of β -adrenergic receptors translates into altered PKA concentration at the sarcomere. We made the choice for the PKA reduction value more clear in the revised manuscript. The main reason for trying to "predict" the changes in cMyBP-C phosphorylation during heart failure was to see if the most well characterized changes (PP1 and PP2A increased, PKA decreased) are sufficient to explain the observed patterns with the model. We thus have not tested further PKA values as there would be little additional insight to be gained in addition to the fitted enzyme concentrations in Figure 4b which also show that there are likely more changes to be expected.

Minor comments:

(i) p.2, 'regulat[i]on'

Authors' response: *Mistake corrected.*

(ii) p.3., second para: Please explain why no other kinases than PKA, RSK2, and PKC (e.g., PKD) were included.

Authors' response: *This was done mostly because the major focus of the study is on the phosphatases and how their integration with kinases determines the overall system response. The chosen enzymes regulate all the four phosphosites that are of interest in this study and because including further kinases would have exceeded the resources available to us. We thus wrote:*

“Although other kinases (e.g. CamKII, PKD) are relevant for cMyBP-C phosphostate regulation, the five enzymes considered here are involved in the regulation of all phosphorylation sites in the m-motif and thus represent a good starting point.”

(iii) p.3, 3rd para and other instances: Please use ‘significant’ (e.g. for slower dephosphorylation) only when supported by a statistical test.

Authors’ response: Mistake corrected.

(iv) Fig. 1d: For later arguments on differences between PP1 and PP2A, one could test for statistical differences of k_{cat} / K_M values.

Authors’ response: We found no statistically significant difference between k_{cat} , K_M and k_{cat}/K_M of PP1 and PP2A for either pS288 or pS308 dephosphorylation. Please see graph below:

Means \pm SEM, $n=3$. Statistical significance between values was assessed by a one-way ANOVA followed by Tukey’s multiple comparison test.

(v) p.6, ‘Allosteric activation of both ...’ – would such activation by specific phosphorylation sites of cMyBP-C be sterically possible?

Authors’ response: Given the presence of the RVxF motif which has been reported as an allosteric regulator and the influence e.g. of many other subunits on PP1 kinetics, we considered the possibility of allosteric regulation plausible and did not perform further structural analyses but decided to directly test this hypothesis on empirical grounds.

(vi) Fig. 3a,b and elsewhere: To compare model predictions and experimental data, please combine them in the same plot (not only a Hill function fit as in Fig. 3b).

Authors’ response:

We now provide a direct comparison of the experimental data and the model prediction (at the exact same enzyme and substrate concentrations as in the experiment) as Supplementary Figure (see **Figure S19a** below). This direct comparison revealed that while the pattern of the dose responses (i.e. shape of the curves and their relative positions) generally matches, the curves are markedly shifted to the left, which is surprising given the quality of the model fits to both PKA and PP1 data.

However, in the PKA / PP1 experiments used for fitting the data, experimental conditions were chosen for the requirements of respective enzymes (needing components such as ATP or Mn^{2+} ions). Since in the dose-response experiment both PKA and PP1 are present, we hypothesized that components required for the activity of one enzyme may hamper the activity of the other. We followed this up by further experiments and indeed found that ATP strongly inhibits PP1 (**Figure S19b**), while $MnCl_2$ strongly reduces activity of PKA (**Figure S19 c**), consistent with earlier reports (cf. <https://doi.org/10.1021/bi970418i>, <https://doi.org/10.1007/BF00223535>, <https://doi.org/10.1002/pro.5560021217>). We conclude that the effective enzyme activities in this experiment do not match with the activities from the experiments used for fitting the model, i.e. the dose-response data from these experimental conditions and the model predictions are not directly comparable in terms of their quantitative features, but only qualitatively. We think that the qualitative agreement of the experimental curves, however, still indicates that the model captures important relationships of cMyBP-C regulation by kinases and phosphatases.

Figure S19: (a) experimental (same data as in Figure 3b of the main text) vs predicted PKA dose-response at 100 nmol/L PP1. (b) pNPP assay in absence and presence of ATP. (c) phosphorylation of 20 μ mol/L cMyBP-C by 2000U PKA in absence and presence of $MnCl_2$.

In future studies, this could be addressed by carefully optimizing the conditions such that both PKA and PP1 simultaneously exhibit high activity or, even better, by collecting data from cell biological or in vivo experiments. However, both options exceed both scope and available resources of the current study, unfortunately.

We accounted for the implications this has for the comparison of the model to human heart failure data (i.e. the results shown in Figure 4) in the discussion and wrote:

“Concerning the interpretation of our results pertaining the human heart data there are two important limitations to consider. Firstly, these predictions rely on the assumption that the optimized in vitro conditions used for the enzymes studied here and the models based on this data can tell us something relevant about the situation in vivo. However, as shown in Figure S19, enzymatic assays can be quite sensitive to a variety of factors. The results on human heart data and the predictions about changes in enzyme activities during heart failure are thus best understood as consistency checks and testable hypotheses about qualitative changes in the signaling activities acting upon cMyBP-C in the failing heart rather than precise quantitative predictions.

(...) This work not only highlights the importance of phosphatases for regulating cMyBP-C phosphorylation, but also provides a useful framework for integrating data on cMyBP-C phosphorylation from cells or animals in future studies.”

(vii) p.12 and Fig. 4c: Please explain in more detail why MAFs are a suitable characterization for the different conditions, and how these are consistent with the patient data (e.g. 'but moderate to strong reductions ...' which are 2P,3P,and 4P).

Authors' response:

We added the following text:

"(...) The purpose of calculating the MAFs is to evaluate under which restrictions (in terms of enzyme concentrations) and for which sites and the phosphorylation states are reduced and under which other conditions these reductions can still be compensated. Thereby, this analysis may help to disentangle which of the observed changes in Figure 4b may be causative and which ones may be compensatory.

*Similar to our dose-response data, we found that the MAFs for intermediate states ($\alpha\beta$, $\alpha\delta$, $\alpha\beta\delta$) are generally lower than for other states. Under HF conditions, the MAFs of most states were unaffected, but moderate to strong reductions were found for $\alpha\beta$, $\alpha\beta\gamma$ and $\alpha\beta\gamma\delta$ (**Figure 4c**, blue vs red). Since these correspond to 2P, 3P and 4P species, this observation is consistent with the strongly reduced amount of 3P and 4P cMyBP-C in the failing donor hearts, indicating that these reductions cannot be sufficiently compensated during HF."*

(viii) p.13 'the effect is likely supported by the sequential ...': Cooperative binding is another potential mechanisms (see Salazar and Hofer above) that could be checked in the parametrized model.

Authors' response:

K_2 is higher than K_5 and K_8 in the final model which is indeed consistent with negative cooperativity, but we see no further relevant difference between K_5 and K_8 . Additionally, these parameters are Michaelis-constants which are of course not the same as dissociation constants. Any rigorous study of this possibility would thus likely require a more in depth analysis including experimental data, exceeding the scope of the current study.

(ix) p.17, 'custom weights': These weights need justification, and a (statistically) cleaner solution would be to omit them.

Authors' response:

The weights were adjusted manually after the automatic calculation of the weights by Copasi resulted in fits with unrealistic transients resulting from the limited time-resolution of the experiments especially at the early phase of the dephosphorylation timecourse (<10 min). However, we followed the referee's suggestion and omitted the weights. The weights can still be found in the Copasi files available in the GitHub repository.

(x) Fig. S21: It is unclear, which predictions the experimental data are compared to (and how the statement in the main text is supported).

Authors' response:

Since the simulations (and experimental data) from this figure were not really relevant to the main conclusions of the paper, we decided to remove the figure and the corresponding main text in order to make the paper more concise.

Reviewer #3 (Remarks to the Author):

In this paper, Kampourakis, Ponnampalani, and Koch describes a mathematical model for the phosphorylation status of cMyBP-C taking both kinases and phosphatases into account, especially PP1 and PP2A. It is important to take all different kinases and phosphatases into account when modelling the reaction kinetics, and the authors do a good job including many of them – but, as they mention themselves, further components also play a role in the cMyBP-C phosphorylation equilibrium. In my opinion, the authors do a good job adjusting/fitting the kinetic model. However, the data the model is based on could be more convincing. The paper is very well written. As I am not an expert in cardiac biology or mathematical modelling, my comments mainly concern the mass spectrometric data.

General points

- To my knowledge nature journals require four replicates for statistical analysis. In most cases the authors report $n=1-3$.

Authors' response: *We have used $n \geq 3-4$ independent repeats for all experiments on which statistical comparisons were performed. We considered lower n -numbers to be appropriate where no statistical comparisons were performed, e.g. the time-courses in Figure S2 were performed with $n=2$, as they were used only for constraining the model and since each reaction was sampled multiple times over a longer duration. Indeed, we find that many recent Nature Communications articles use less than four replicates (see e.g. <https://www.nature.com/articles/s41467-024-44932-w>).*

- The multi-phosphorylated peptide is essential for the story. The authors reference a previous study for the generation of this, but including a short description on how it was made (and how the phosphorylation site specificity checked) would be valuable.

Authors' response: *We have added a detailed description and associated references of how the phosphor-peptides were prepared to the manuscript (please see supplemental information). The PKCepsilon site (delta site) was previously identified and validated in rodent cardiac myosin binding protein-C (please see Xiao et al., 2007, Biochemistry).*

- The concept of making kinetic models taking several kinases and phosphatases into account is not trivial. Having excellent data to base the models on is a necessity. The data presented in this paper could be more convincing. One idea could be to analyze the phosphorylated peptide using mass spectrometry, making sure all four phosphosites are identified as a start. The peptide could then be incubated with the different phosphatases, or a combination, and samples taken for mass spectrometric analysis at different time points – and maybe normalized to the intensity/abundance of the individual sites based on the control sample (with all four sites). And, all of this performed in quadruplicates of course to ensure statistical significance.

Authors' response: *We thank reviewer for this criticism but disagree that our data are not convincing (please also see reviewer #1 and #2, who clearly indicate that data we present is of high quality).*

The methodological focus of the paper is on cardiac biochemistry and computational modelling. Experimentally, we used a wide range of protein biochemical and biophysical methods (e.g. Phostag-SDS-PAGE, Western blotting, mass spectrometry, enzyme kinetics, fluorescent spectroscopy, pull-down experiments, microscale thermophoresis) to study and

validate the phosphorylation/dephosphorylation mechanism of cMyBP-C. The mass spectrometry data constitutes only a fraction of this data and was used to confirm the mechanism that was elucidated by other methods. Moreover, we are reliant on external mass spectrometry services and time-resolved experiments with n-numbers of 4 as suggested by the reviewer would exceed our budget by far.

However, we have additionally obtained site-specific antibodies against the main phosphorylation sites in cMyBP-C (please see new Figure 1c), and following the dephosphorylation kinetics of cMyBP-C using these antibodies supports the originally proposed dephosphorylation mechanism.

- While I appreciate that the authors developed the first integrated model of cMyBP-C phosphorylation as a function of multiple kinases and phosphatases, it would be a good addition to the story, if they could speculate on how this could be used or would be beneficial in the future.

Authors' response: We thank the referee for this suggestion. We added the following paragraph to the discussion:

“Our study not only highlights the importance of phosphatases for regulating cMyBP-C phosphorylation, but also provides a useful framework for integrating data on cMyBP-C phosphorylation from cells or animals in future studies. Modulating cMyBP-C phosphorylation is increasingly considered as a potentially promising therapeutic strategy⁶⁰⁻⁶³. An integrated and validated model of how cMyBP-C is regulated by both phosphatases and kinases could be used to design therapeutic strategies to optimize cMyBP-C phosphorylation in heart disease by modulating multiple rather than single enzymatic activities. Such a strategy could not only prove more effective, but may require lower doses than single-target therapies. Since cMyBP-C is a hub for pathogenic mutations, it further would be interesting to study the effect of relevant mutations on (de)phosphorylation and how physiological cMyBP-C phosphorylation patterns may be restored.

Comments to specific pages

- Page 3: The authors write: “Physiologically, the interaction of the cMyBP-C with both filament systems is controlled by phosphorylation at several sites in the cardiac-specific m-motif by physiologically relevant protein kinases”. Maybe change to “Physiologically, the interaction of the cMyBP-C with both filament systems is controlled by phosphorylation at several sites in the cardiac-specific m-motif by protein kinases”.

Authors' response: Done.

- Page 3: For the sentence above, maybe the references should be added before the reference to the figure?

Authors' response: Done.

- Page 4: The authors use a 50% mixture of the peptide containing 3 or 4 phosphorylation sites. Why is this necessary? Is it in order to be able to distinguish them using PhosTag?

Authors' response: We used a mixture of tris- and tetrakis – phosphorylated C1mC2 to follow their relative dephosphorylation kinetics in more detail using Phostag technology. It is not strictly necessary to use a mixture (and we have also used homogenous mono-, bis- and tris-

phosphorylated substrates in the Figure S2), but we considered this slightly more complex experimental design to be an interesting case for the subsequent model fitting/testing.

- Page 4: The authors claim that because they only identify pS279 and pS288 using mass spectrometry, pS308 and pS313 are likely dephosphorylated first. I am not convinced by this. Absence of evidence is not equal evidence of absence. The experiment lacks a proper control and replicates (see points regarding mass spectrometry data below).

Authors' response: *We now include additional Western-blot experiments using phospho-specific antibodies which further corroborate that pS308 is dephosphorylated fast and at any point, followed by pS279 while the pS288 is still unchanged (Figure 1c of the revised manuscript). Moreover, the Michaelis-Menten kinetics (already shown in the previous version of the manuscript), too, clearly indicate that pS288-C1mC2 is a far poorer substrate than pS308. Together, this supports the proposed reaction sequence. The presence of mainly pS288 and pS279 in the mass spectrometry data is consistent with this as are the timecourses for pS288, pS279-C1mC2 and for pS288-pS308-C1mC2 shown in Figure S2, where after a rapid consumption of the 2P species, only a 1P substrate remains.*

- Page 5, figure 1c: The authors claim that because the pS288 signal is relatively stable, and the signal correlates with that for the mono-phosphorylated species, that pS288 likely accounts for the slow dephosphorylation phase. This is a rather strong conclusion, and would benefit from more evidence. The authors could for example consider including a positive control by including a condition where a phosphatase capable of removing phosphorylation on pS288 was included (if one exists?). This would also serve as a control for the antibody.

Authors' response: *Sufficient quantitative evidence for this claim comes in our opinion from the Michaelis-Menten data shown in Figure 1d. The estimated K_m values of approx. 200 μM show that pS288-C1mC2 really is a poor substrate for PP1/PP2A, i.e. that the reaction only slowly. In contrast, the K_m values for pS308-C1mC2 are much lower (about 44 μM), i.e. the reaction is faster (since k_{cat} values are similar).*

However, as described in our response to the referee's previous comment, the additional Western-blot experiments further support the proposed model of slow pS288 phosphorylation.

- Page 5, figure 1d: This figure is not clear to me. To me it looks like there is more of the dephosphorylated form when treating with lower concentrations of the phosphatase? I also do not see a clear difference between the two blots/substrates.

Authors' response: *This may be confusing because we loaded equal protein amount to each lane in the Phostag gels (to make sure signals are of comparable intensity in each sample). However, each sample had different substrate concentrations. Therefore it can appear that at higher substrate concentration there is less conversion. However, the fraction of conversion needs to be multiplied by the substrate concentration to get the molar conversion per time unit, yielding the Michaelis-Menten relationship shown in Figure 1d.*

In the revised methods, we now mention that we loaded equal amounts of protein to each lane.

- Page 6: It would be great if the authors could add more experimental evidence to support their proposed reaction mechanism for cMyBP-C dephosphorylation.

Authors' response: We now include additional Western-blot experiments using phospho-specific antibodies (Figure 1c in the revised manuscript) which further corroborate the proposed reaction mechanism.

• Page 6, figure S1: It is great that the authors include additional data for the various variants of phosphorylation. However, it would be great if the authors could specify how they generated the unique phospho-versions, and check that it is actually only this species present.

Authors' response: Only single bands in appeared in the Phostag gel and all constructs were checked with electron spray ionisation mass spectrometry to confirm a homogeneity. We have added this information to the methods section.

• Page 11, figure 3: I understand that it can be difficult to include many things in one graph, but the 3D graphs are not clear to me.

Authors' response: We thank the referee for pointing out the difficulty of the plots. The 3D plots show dose-response relationships (response in terms of cMyBP-C phosphorylation) for increasing PKA concentrations at fixed phosphatase concentration. Essentially, each of the shown slices corresponds to a dose-response at a different phosphatase concentration:

To make this clearer we added the following explanation highlighted in bold to the legend of Figure 3a:

“(…) a, predicted steady-state cMyBP-C phosphorylation response to increasing PKA concentrations and at different concentrations of PP1 (**each slice corresponds to a different, fixed phosphatase concentration**).(...)”

Points regarding the mass spectrometry data

- The authors include two mass spectrometric RAW files. To account for biological and technical variations, replicates are needed.
- It was not trivial downloading the RAW files from the MassIVE repertoire. In my experience, ProteomeXchange is preferred for uploading MS data.
- The methods available through MassIVE are missing some essential details. It states that

“enriched phosphopeptides were loaded...”, but how were the phosphopeptides enriched? Details regarding the gradient on the LC and the settings on the MS are also missing.

- The authors/mass spec facility search the data using Peaks. To my knowledge, this software is not the best for localizing PTMs. Search settings are also not described.
- I tried researching the two RAW files using MaxQuant (free software) with cMyBP as a FASTA file, and phosphorylation on serine, threonine, and tyrosine as variable modifications. From this search, pS279 is the most intense/abundant site, and pS288 the second most intense/abundant. A small fraction of the intensity was observed on pS290, and even less on pS312. As MaxQuant reports the localization probability (how likely is it that the modification is actually residing on this reported amino acid residue), it should be noted, that the localization probability of pS288 is only 0.63, whereas class I phosphosites have a localization probability above 0.75. This is not necessarily a problem, if the spectrum looks good (I did not check this).
- In order to conclude anything from mass spectrometric data, a control sample with all four phosphorylation sites identified (and localized) is needed. The two sites not found by mass spectrometry (pS308 and pS313) reside in a sequence surrounded by lysines and arginines, making the trypsin-digested peptides likely too short to identify as peptides often need to be minimum 7 amino acids long. This could potentially be overcome by using different proteases. But in order to use mass spectrometry to proof which phosphosites are present, the phosphosites identified under phosphatase treated conditions should be compared directly to phosphosites in an untreated sample.

Authors' response:

We have chosen the MassIVE repository as it was recommended to us by the service provider that conducted the MS analysis for us. It is fully compliant with requirements of Nature Communications. We also now added a full method description for the MS.

Regarding PEAKS: the service provider considers PEAKS a good choice and pointed out other publications on this topic:

Han X, Xin L, Shan B, Ma B. PeaksPTM: Mass spectrometry-based identification of peptides with unspecified modifications. J. Proteome Res., 10, 2930-2936 (2011). <https://pubs.acs.org/doi/10.1021/pr200153k>

Wan N, Wang N, Yu S, et al. Cyclic immonium ion of lactyllysine reveals widespread lactylation in the human proteome. Nat Methods. 2022;19(7):854-864. doi:10.1038/s41592-022-01523-1

We also now provide the search parameters for PEAKS in the methods.

According to the York Proteomics Lab, using FragPipe (an other free software) similar results are obtained. However, the most intense signal is for pS288 with a best site localisation probability of 0.7696 – just above the reviewer's suggested threshold of 0.75. That said, all thresholds are somewhat subjective, the most important thing is that the level of ambiguity is acceptable to the usage.

Spectral annotations are provided below from the original PEAKS search.

Site localisations for these positions are not highly confident because of adjacent phosphorylatable residues.

It is practically very hard to resolve the specificity of adjacent residues – i.e here the phosphorylation can't really be resolved between the adjacent Thr and Ser residues from the MS data alone. In this case with such similarly located positions the localisation probabilities will never be high but if you are confident only Ser will be affected biologically or the ambiguity of the adjacent Thr/Ser is not important than it is not an issue.

Lastly, while pS308 and pS313 reside in a sequence surrounded by lysines and arginines, we previously (Ponnam et al. 2019, <https://www.pnas.org/doi/full/10.1073/pnas.1903033116>) were able to identify without problem them in a fully phosphorylated version of the same substrate using the same protease, the same service provider (York Proteomics Lab) and the same kinases. We thus do not think that the absence of pS313 and pS308 is due to missed peptide identification. However, while we think the MS data as they currently are still provide value for the confirmation of enzymatic mechanism, we appreciate that these alone are not sufficient to conclude the full dephosphorylation mechanism. Yet, we are currently unable to conduct further MS experiments due to budget reasons for this project. As outlined above, we think that the PhosTag timecourse data, the enzyme kinetics and the newly added Western-Blotting data are together sufficient to determine the site-specificity of C1mC2 dephosphorylation by PP1 and PP2A and conclusions of the presented work.

Reviewers' Comments:

Reviewer #1:

Remarks to the Author:

The authors have satisfactorily addressed my concerns.

Reviewer #3:

Remarks to the Author:

The authors responded well to the reviewer comments, and included relevant new data. However, I still have a few concerns.

The authors prepared phospho-specific antibodies for three of the phosphorylation sites, but should describe how these were generated in the methods section, as well as showing the validation of them.

The authors reply to reviewer 1, that they have added additional mass spec data. Where is this shown?

Can the authors clarify how they know that pS313 is also dephosphorylated first (p4 l152)?

I still believe that the story would have greatly benefitted from additional mass spec data (control with all four sites present and then the time point with two sites present performed using the same settings), but I also understand that it is a minor part of the story - and supports the other findings.

Reviewer #4:

Remarks to the Author:

Thank you for the thorough and considerate responses to each of the concerns and comments. The detailed answers and changes made to the manuscript have significantly improved the quality of this work, and any changes not made based on the reviewer comments were adequately justified. This reviewer finds that the modeling elements of the study provide valuable new insights into mechanisms of phosphatase regulation of cMyBP-C.

Reviewer #3 (Remarks to the Author):

The authors responded well to the reviewer comments, and included relevant new data. However, I still have a few concerns.

The authors prepared phospho-specific antibodies for three of the phosphorylation sites, but should describe how these were generated in the methods section, as well as showing the validation of them.

Authors' response: *We have included a brief description of the process in the methods and included the validation data (ELISA from the company, WB from us).*

The authors reply to reviewer 1, that they have added additional mass spec data. Where is this shown?

Authors' response: *We have indeed not shown additional mass spec data. We first added the mass spec data from Ponnam et al. 2019 to address reviewer 1's request on peptide identification but then changed our mind as we considered it inappropriate and unnecessary to republish the same data. Instead we only referred to this paper but forgot to change our response. We apologize for the confusion this has caused.*

Can the authors clarify how they know that pS313 is also dephosphorylated first (p4 l152)?

I still believe that the story would have greatly benefitted from additional mass spec data (control with all four sites present and then the time point with two sites present performed using the same settings), but I also understand that it is a minor part of the story - and supports the other findings.

Authors' response: *This is essentially the same question as the referee's previous question:*

“Page 4: The authors claim that because they only identify pS279 and pS288 using mass spectrometry, pS308 and pS313 are likely dephosphorylated first. I am not convinced by this. Absence of evidence is not equal evidence of absence. The experiment lacks a proper control and replicates (see points regarding mass spectrometry data below). “

To paraphrase our previous answer to this question in more detail: the reaction scheme shown in Fig 1e (in which there's a sequential mechanism for the sites pS313→pS279→pS288 and where pS308 can be dephosphorylated at any point) is the best description (in our opinion) which we concluded from the joint consideration of all the experiments (Michaelis-Menten kinetics, Phostag time courses, Western-blotting and mass spectrometry). Starting with rapid pS308 removal at any point: this is supported by the findings that pS308 can be rapidly dephosphorylated from monophosphorylated pS308-C1mC2 (Fig 1d) but also from pS313,pS308,pS288,pS279/ pS308,pS288,pS279-C1mC2 (Fig 1c) and does not show up in the mass-spec analysis of bisphosphorylated C1mC2 from the dephosphorylation reaction (Suppl. Fig 1). Next, that pS288 is very likely the site to be dephosphorylated last is supported both by the slow dephosphorylation kinetics of monophosphorylated pS288-C1mC2 (Fig 1d) and the WB time courses (Fig. 1c). Given this slow dephosphorylation, that pS279 is dephosphorylated before pS288 would also be consistent with the Phostag dephosphorylation time courses of pS288,pS279-C1mC2 from Suppl. Fig. 2, where after a rapid decline of pS288,pS279-C1mC2, a monophosphorylated species remains that only decreases very

slowly in concentration. Dephosphorylation of pS279 before pS288 is also shown by the WB time courses (Fig. 1c). The only question open to conclude the scheme shown in Fig. 1e now remains: is pS279 or pS313 dephosphorylated first from pS313, pS288, pS279-C1mC2 or may they even be phosphorylated randomly, i.e. via the sequence pS313,pS288,pS279→pS288,pS279 and pS313,pS288,pS279→pS313,pS288? The mass-spec analysis of bisphosphorylated C1mC2 from the dephosphorylation reaction (Suppl Fig 1) in which pS288 and pS279 are the predominant signal for both PP1 and PP2A suggests that indeed the sequence is not random and that pS313 is dephosphorylated before pS279. Although the referee has in her/his previous report raised the concern that pS313 may not be properly identified due to technical reasons (“[...] The two sites not found by mass spectrometry (pS308 and pS313) reside in a sequence surrounded by lysines and arginines, making the trypsin-digested peptides likely too short to identify as peptides often need to be minimum 7 amino acids long [...]”), we were able to identify pS313-containing peptides clearly and without issues from the same C1mC2 fragment using the same method (i.e. trypsin digestion and analysis done by the same mass-spec service provider) in Ponnam et al., 2019 PNAS as we also mentioned in our previous reply. We acknowledge that a more detailed mass spectrometric analysis would have provided additional value. However, we also do not claim that the proposed reaction scheme from Fig 1e is a final truth beyond all doubt and simply think of it as the best description of the data so far and have chosen our wording carefully in the text and figure legend:

This **suggests** that in the fully phosphorylated C1mC2 fragment, pS308 and pS313 are dephosphorylated first, leaving only pS279 and pS288.

(...) e, **proposed** reaction scheme for dephosphorylation of cMyBP-C by PP1 and PP2A. (...)

In addition to the direct data on the dephosphorylation reaction scheme provided in our study, we would like to mention three further points which increase the plausibility of the proposed scheme: First, the forward reactions catalysed by PKA in absence of phosphatases was shown to follow strictly the sequence pS288→pS279→pS313 (Ponnam et al. 2019). If this sequence has a specific biological relevance/function, it makes sense that proposed dephosphorylation scheme is exactly in the reverse order, as otherwise the phospho-state-distribution in the simultaneous presence of kinases and phosphatases (as it is the case in vivo) would deviate from this sequential order. Secondly, we also provide an explanation of what the structural mechanism underlying the sequential phosphorylation and proposed sequential dephosphorylation is. The proposed principle of the reaction mechanism being based on the substrate confirmation further has the advantage that it makes the phosphorylation sequence independent of the specific enzymes. Lastly, in agreement with the proposed sequence pS288→pS279→pS313, the fragments pS313,pS288,pS279-, pS288,pS279- and pS288-C1mC2 not only have clearly distinct signals in the SYPRO-assay (which we think indicates different conformational states in agreement with the idea that the conformation determines the sequence), but also show different functional effects e.g. on actomyosin ATPase (cf. Fig 5, Ponnam et al. 2019).